# Do-PFN: In-Context Learning for Causal Effect Estimation

**Jake Robertson**[1,2,3*]    **Arik Reuter**[4,5,*]    **Siyuan Guo**[4,5,1]    **Noah Hollmann**[1]
**Frank Hutter**[1,2,3,†]    **Bernhard Schölkopf**[2,4,†]

[1]Prior Labs, Freiburg, Germany
[2]ELLIS Institute Tübingen, Tübingen, Germany
[3]University of Freiburg, Freiburg, Germany
[4]Max Planck Institute for Intelligent Systems, Tübingen, Germany
[5]University of Cambridge, Cambridge, United Kingdom

[*]Equal contribution    [†]Equal supervision

## Abstract

Causal effect estimation is critical to a range of scientific disciplines. Existing methods for this task either require interventional data, knowledge about the ground-truth causal graph, or rely on assumptions such as unconfoundedness, restricting their applicability in real-world settings. In the domain of tabular machine learning, Prior-data fitted networks (PFNs) have achieved state-of-the-art predictive performance, having been pre-trained on synthetic causal data to solve tabular prediction problems via in-context learning. To assess whether this can be transferred to the problem of causal effect estimation, we pre-train PFNs on synthetic data drawn from a wide variety of causal structures, including interventions, to predict interventional outcomes given observational data. Through extensive experiments in synthetic and semi-synthetic settings, we show that our approach allows for the accurate estimation of causal effects without knowledge of the underlying causal graph.

## 1 Introduction

The estimation of causal effects is fundamental to scientific disciplines such as medicine, economics, and the social sciences (Pearl, 2009; Varian, 2016; Imbens, 2024; Wu et al., 2024). Questions such as "Does a new drug reduce the risk of cancer?" and "What is the impact of minimum wage on employment?" can only be answered by taking the causal nature of the problem into account.

The widely accepted gold standard for assessing causal effects are randomized controlled trials (RCTs). While RCTs allow for the direct estimation of causal effects, they can sometimes be unethical or expensive, and, in many cases, simply impossible. In contrast to experimental data from RCTs, *observational* data is more accessible, collected without interfering in the independent and identically distributed (i.i.d) data-generating process. Estimating causal effects from observational data alone can be challenging or even impossible without strict assumptions (Spirtes et al., 1993).

Various methods have been proposed to address the problem of causal effect estimation, typically relying on the assumption of unconfoundedness (Rosenbaum and Rubin, 1983). This assumption states that, conditional on a set of observed covariates, treatment assignment is independent of the potential outcomes. While this condition enables identification of causal effects from observational data, it can be difficult to verify or justify in practice, as it requires that relevant confounders are observed and properly accounted for (Hernán and Robins, 2010; Imbens and Rubin, 2015). Under unconfoundedness, a variety of estimation techniques have been developed, including causal forests

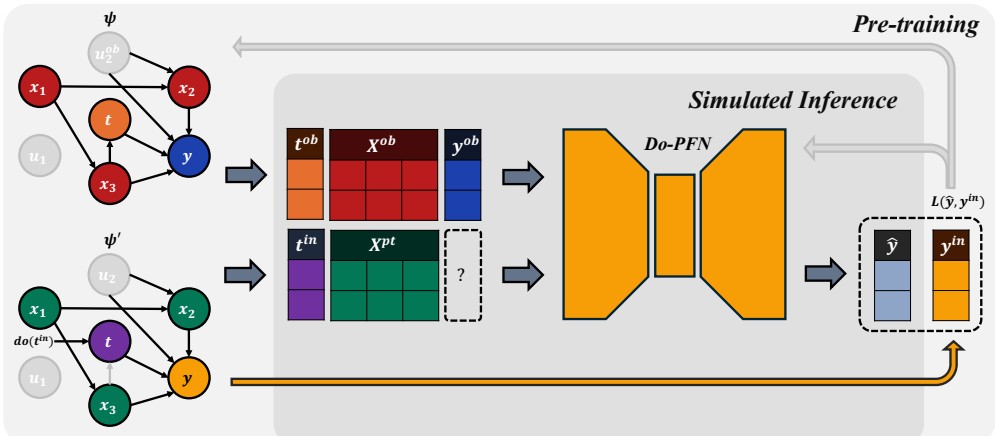

Figure 1: **Do-PFN overview**: Do-PFN performs in-context learning (ICL) for causal effect estimation, predicting conditional interventional distributions (CIDs) based on observational data alone. In pre-training, a large number of structural causal models (SCMs) is sampled. For each SCM, we sample an entire dataset of $M^{ob}$ *observational* data points $\mathcal{D}^{ob} = \{(t_j^{ob}, \mathbf{x}_j^{ob}, y_j^{ob})\}_{j=1}^{M^{ob}}$. We also sample $M^{in}$ *interventional* data points $\mathcal{D}^{in} = \{(t_k^{in}, \mathbf{x}_k^{in}, y_k^{in})\}_{k=1}^{M^{in}}$. To simulate inference, we input $(t^{in}, x^{in})$ along with the entire observational dataset $\mathcal{D}_{ob}$, which can have various sizes and dimensionalities. Subsequently, the transformer makes predictions $\hat{y}$, and we calculate the pre-training loss $L(\hat{y}, y^{in})$ between the predictions $\hat{y}$ and the ground truth interventional outcomes $y^{in}$. Pre-training repeats this procedure across millions of sampled SCMs to *meta-learn* how to perform causal inference *in context*. In applications, Do-PFN leverages the many simulated interventions it has seen during pre-training to predict CIDs, relying only on observational data and requiring no information about the causal graph.

(Wager and Athey, 2018), or "doubly robust" methods that combine propensity score modeling and outcome regression (Chernozhukov et al., 2018).

Many applications of causality involve tabular data. Prior-data fitted networks (PFNs) (Müller et al., 2022) have recently transformed the landscape of tabular machine learning. TabPFN (Hollmann et al., 2023, 2025), an application of PFNs to tabular classification tasks, was initially met with skepticism, arguably because of its radically different working principle compared to other state-of-the-art tabular machine learning methods; in a nutshell, it is a model that takes as input an entire dataset of labeled training data points $\{(\mathbf{x}_i, y_i)\}_{i=1}^M$ and unlabeled test data points $\boldsymbol{x}_{\text{new}}$ and "completes" it with an *in-context* prediction of the corresponding $y_{\text{new}}$, akin to a large language model answering a question by producing a likely text continuation. The approach can handle datasets of various sizes and dimensionality (in which case matrices get zero-padded). TabPFN is trained to execute this completion task across millions of synthetic datasets, for each dataset computing the likelihood of the true $y_{\text{new}}$ under the PFN's predictive distribution and performing one step of stochastic gradient descent on this likelihood. The *context* refers to the chunk of data that is fed into the model at a time. In a language model, these are thousands of words; in the case of TabPFN it is a whole training set plus a query data point; and in our case also an added intervention. *In-context learning* refers to the ability to output a desired quantity based on what is provided in the context; the term *learning* indicates that this solves a task that would classically often require learning (e.g., estimating a regression function and using it to predict a target). For us, the desired quantity is the effect of an intervention. Training such a model then consists of *meta-learning* (across millions of meta-training datasets) the ability to provide the desired answer given a context. The term *meta* indicates that it is an outer loop around a procedure that itself is already a form of learning or estimation.

In spite of being pre-trained only on synthetic data, TabPFN has produced impressive results on real-world machine learning benchmarks (McElfresh et al., 2023; Xu et al., 2025; Hollmann et al., 2025). Given these remarkable findings, it is timely to assess whether a similar meta-learning approach could help us tackle harder problems that are causal rather than merely predictive. Due to the delicate nature of causal inference, the sensitivity of the tasks associated with it, and the scarcity of real-world causal effect estimation data with ground truth, we will do so by systematically exploring synthetic data with known ground truth.

Recent developments have shown that certain limitations in inferring causal structures and causal effects can be addressed by using multi-domain data in the form of mixtures of i.i.d. observational data (Guo et al., 2023, 2024). Interestingly, PFNs also leverage pre-training on a mixture of i.i.d. data to meta-learn how to solve predictive tasks at test time. We thus hypothesize that some causal tasks at test time could also be addressed through meta-learning on multi-domain data. As a first step, our goal is to extend PFNs to the problem of estimating conditional interventional distributions (CIDs).

In contrast to TabPFN, we not only simulate observational tabular data in order to predict a target feature. Rather, we additionally simulate causal interventions, teaching our model, which we call *Do-PFN*, to meta-learn how to perform causal inference.

**Our contributions**

1. **Do-PFN**: We propose Do-PFN, a foundation model pre-trained on data from structural causal models (SCMs) that can predict interventional outcomes and causal effects from observational data.

2. **Semi-synthetic evaluation**: We evaluate the performance of Do-PFN on six case studies across more than 1,000 synthetic datasets, the popular RealCause benchmark (Neal et al., 2020), as well as two observational datasets with widely agreed upon causal graphs. We provide ablation studies within our prior, an out-of-distribution analysis, assess uncertainty calibration, and evaluate Do-PFN against a competitive set of meta-learners, doubly robust, and deep-learning-based methods in the task of CATE estimation.

3. **Theoretical results**: In providing a mathematical underpinning for Do-PFN, we prove that it can achieve an optimal approximation of the conditional intervention distribution (CID) concerning the chosen prior over data-generating functions. We also provide a characterization of the sources of uncertainty in our model, and present a consistency argument to show which types of uncertainty vanish with infinite data.

## 2   Background and related work

**Structural causal models**   Structural causal models (SCMs; Pearl, 2009; Peters et al., 2017) represent the structure of a data-generating process. The first component of an SCM $\psi$ is a directed acyclic graph (DAG) $\mathcal{G}_\psi$, which we assume to have $K$ nodes, each representing a variable $z_k$. Furthermore, the SCM specifies the mechanisms to generate the variables from their (causal) parents via structural equations $z_k = f_k(z_{\mathrm{PA}(k)}, \epsilon_k)$, where $f_k$ is a function, $z_{\mathrm{PA}(k)}$ denotes the parents of variable $k$ in $\mathcal{G}$ and $\epsilon_k$ is a random noise variable. We use $\epsilon := (\epsilon_1, \epsilon_2, \ldots, \epsilon_K)$ to denote the vector comprising the noise terms. In our simulations these will be taken as jointly independent, but our methodoldy does not require this.

**Interventions and causal effects**   In the context of SCMs, performing an intervention $do(t)$ for a binary variable $T \in \{z_1, z_2, \ldots, z_K\}$ that is part of the SCM $\psi$ corresponds to removing all incoming edges into the node representing $t$ and fixing the value of the variable $T$ to the value $t$. We assume the "treatment" $T$ to be binary such that $t \in \{0, 1\}$. The causal effect of this intervention on an outcome $y$ is captured by $p(y|do(t), \psi)$.

A central object of interest for this paper is the conditional interventional distribution (**CID**; Shpitser and Pearl, 2006) that additionally conditions on a vector $\mathbf{x}$ comprising several variables in the SCM,

$$p(y|do(t), \mathbf{x}). \tag{1}$$

A CID answers a question like *"What is the distribution of outcomes given that (i) a patient has features $\mathbf{x}$ and (ii) an intervention $do(t)$ is performed?"* CIDs enable the estimation of conditional average treatment effects (**CATE**s): $\tau(x) := \mathbb{E}[y|do(1), \mathbf{x}] - \mathbb{E}[y|do(0), \mathbf{x}]$.

**Estimating causal effects**   Various methods allow for the direct estimation of causal effects from experimental data (Shalit et al., 2017; Kennedy, 2023; Nie and Wager, 2021). However, RCT data is often difficult to access. It might be easier, or even the only option, to access an *observational* dataset $\mathcal{D}_{ob} = \{(y_j^{ob}, t_j^{ob}, x_j^{ob})\}_{j=1}^{M_{ob}}$ of passively collected samples $(y_j^{ob}, t_j^{ob}, x_j^{ob}) \sim p(y, t, \mathbf{x})$.

When approaching causal effect estimation from the framework of *do-calculus* (Pearl, 2009), practitioners first need to construct an SCM $\psi$ that they believe (or have inferred) to represent the

ground-truth data-generating process. The rules of do-calculus subsequently allow to determine whether and how the desired causal effect can be estimated from the data. Back-door and front-door adjustment are popular methods to allow for estimation of the desired causal effects.

The Neyman-Rubin framework (Imbens and Rubin, 2015) defines causal effects as contrasts between potential outcomes $y_1 \sim p(y|do(1))$ and $y_0 \sim p(y|do(0))$, and relies on a set of key assumptions, critically ignorability (or unconfoundedness), which requires that treatment assignment is independent of potential outcomes given a set of observed covariates. Machine-learning based methods that are conceptualized in this framework include causal trees (Athey and Imbens, 2016), causal forests (Wager and Athey, 2018), as well as T-, S- and X-learners (Künzel et al., 2019).

**Prior-data fitted networks and amortized Bayesian inference** In our context, we define amortized (Bayesian) inference as learning the mapping $\mathcal{D} \mapsto p(y|\mathbf{x}, \mathcal{D})$ from a dataset to a posterior; that is, the amortization occurs at the dataset level. The model that parameterizes this mapping can be obtained by simulating a large number of samples of the form $(\mathcal{D}_i, y_i, \mathbf{x}_i)$, followed by training the model to predict $y_i$ while conditioning on $\mathbf{x}_i$ and $\mathcal{D}_i$. Neural processes (Garnelo et al., 2018a,b; Nguyen and Grover, 2022) and various techniques from the field of simulation-based inference (Wildberger et al., 2023; Gloeckler et al., 2024; Vasist et al., 2023) perform amortized inference in the aforementioned manner. Recently, PFNs have been proposed as an amortized inference framework, emphasizing the role of large-scale pre-training and realistic simulators of synthetic data, referred to as the *prior* (Müller et al., 2022). The PFN framework has been successfully applied to diverse problems, such as time-series prediction (Dooley et al., 2023; Hoo et al., 2024), Bayesian optimization (Müller et al., 2023; Rakotoarison et al., 2024), and counterfactual fairness (Robertson et al., 2024).

**Amortized causal inference** Amortized inference beyond observational distributions was first explored for causal discovery (Ke et al., 2022; Lorch et al., 2022; Dhir et al., 2025). Sauter et al. (2025) consider the problem of meta-learning causal inference, proposing to learn the shift in distributions of all nodes in the SCM when performing an intervention. However, this approach fails to outperform a conditioning-based baseline even in a two-variable setting. Nilforoshan et al. (2023) consider the problem of zero-shot inference for CATEs using a meta-learning approach to facilitate generalization to unseen treatments. Concurrent to our work, Bynum et al. (2025) propose to use amortized inference to learn various causal effects; however, they only focus on low-dimensional SCMs with up to three nodes and do not target the CID, but only point estimates, thus ignoring uncertainty.

## 3    Methodology: causal inference with PFNs

**Modeling assumptions** We now formalize how to do causal inference with PFNs, more precisely how to estimate conditional interventional distributions (CIDs) defined as $p(y|do(t), \mathbf{x})$ from observational data $\mathcal{D}^{ob}$. A central component of our approach to causal effect estimation is to posit a prior $p(\psi)$ over SCMs. We further require that every sampled SCM $\psi \sim p(\psi)$ allows to simulate observational data from $p(y^{ob}, t^{ob}, \mathbf{x}^{ob}|\psi)$ by sampling noise $\epsilon \sim p(\epsilon)$ that is propagated through the SCMs $\psi$. We furthermore assume a prior $p(t^{in})$ over possible values for the treatment variable when performing an intervention $do(t^{in})$. This prior is only required to sample values for the intervention and does not affect how we model the CID (Equation 1) provided it has sufficient support. Samples from the distribution $p(y^{in}, \mathbf{x}^{in}|\psi, do(t^{in}))$ over outcomes and covariates given this intervention then result from forward-propagating through the intervened-upon SCM. Please refer to Algorithm 1 and Appendix C for more details on the data-generating process.

The assumptions above imply the following form of the CID:

$$p(y^{in}|do(t^{in}), \mathbf{x}^{in}) = \int p(y^{in}|do(t^{in}), \mathbf{x}^{in}, \psi)\, p(\psi|\mathbf{x}^{in})d\psi.^1 \tag{2}$$

Assuming a prior $p(\psi)$ over SCMs, and thus also over causal graphs $\mathcal{G}_\psi$, can be seen as an extension of the classical do-calculus approach where typically a fixed causal graph $\widetilde{\mathcal{G}_\psi}$, or even a fixed SCM $\widetilde{\psi}$, is used as the basis for further inference. Compared to the assumptions typically made in the potential outcomes framework, our method also includes scenarios without the unconfoundedness assumption.

---

[1] Note that in our framework $p(\psi|\mathbf{x}^{in}) \neq p(\psi)$ since knowing the feature vector $\mathbf{x}^{in}$ provides information on the SCM that generated it.

**Algorithm 1:** Prior-fitting with SGD. Do-PFN is pre-trained on pairs of synthetic observational and interventional datasets; the model is trained to predict interventional outcomes $y^{in}$ given a covariate-vector $\mathbf{x}^{in}$, the value of an intervention $t^{in}$ and an observational dataset $\mathcal{D}^{ob}$.

```
 1  for i = 1, 2, ..., N do
 2  |    Draw ψᵢ ~ p(ψ); // Draw an SCM
 3  |    Initialize 𝒟ᵢᵒᵇ ← ∅;
 4  |    Draw M_ob ~ Uniform({M_min, M_min + 1, ..., M_max}); // Number of
    |        observational data points
 5  |    for j = 1, ..., M_ob do
 6  |    |    Sample noise εⱼ ~ p(ε);
 7  |    |    Draw yⱼᵒᵇ, tⱼᵒᵇ, 𝐱ⱼᵒᵇ ~ p(yᵒᵇ, tᵒᵇ, 𝐱ᵒᵇ|ψᵢ, εⱼ); // Draw observational data
 8  |    |    𝒟ᵢᵒᵇ ← 𝒟ᵢᵒᵇ ∪ {(yⱼᵒᵇ, tⱼᵒᵇ, 𝐱ⱼᵒᵇ)};
 9  |    end
10  |    Initialize 𝒟ᵢⁱⁿ ← ∅;
11  |    Set M_in = M_max − M_ob;
12  |    for k = 1, 2, ..., M_in do
13  |    |    Sample noise εₖ ~ p(ε);
14  |    |    Draw 𝐱ₖⁱⁿ ~ p(𝐱ⁱⁿ|ψᵢ, εₖ); // Pre-treatment values of covariates
15  |    |    Draw tₖⁱⁿ ~ p(tⁱⁿ); // Draw value for intervention
16  |    |    Draw yₖⁱⁿ ~ p(yⁱⁿ|do(tₖⁱⁿ), ψᵢ, εₖ); // Sample interventional outcomes
17  |    |    𝒟ᵢⁱⁿ ← 𝒟ᵢⁱⁿ ∪ {(yₖⁱⁿ, tₖⁱⁿ, 𝐱ₖⁱⁿ)};
18  |    end
19  |    Compute ℒᵢ(θ) = ∑_{k=1}^{M_in} − log q_θ(yₖⁱⁿ|do(tₖⁱⁿ), 𝐱ₖⁱⁿ, 𝒟ᵢᵒᵇ); // Loss computation
20  |    θ ← θ − α∇ℒᵢ(θ); // Gradient descent
21  end
```

**Approximating the conditional interventional distribution** Ultimately, we are interested in obtaining a model $q_\theta(y^{in}|do(t^{in}), \mathbf{x}^{in}, \mathcal{D}^{ob})^2$ that is as close as possible to the CID $p(y^{in}|do(t^{in}), \mathbf{x}^{in}, \psi)$ for all relevant treatment values $t$, SCMs $\psi$, and covariate-vectors $\mathbf{x}^{in}$, while only taking observational data $\mathcal{D}^{ob}$ into account. The core idea of PFNs is to achieve this by *prior fitting*, i.e., minimizing the negative log-likelihood $-\log q_\theta(y^{in}|do(t^{in}), \mathbf{x}^{in}, \mathcal{D}^{ob})$ on data from the synthetic data-generating process (Müller et al., 2022) via stochastic gradient descent (lines 19 and 20 in Algorithm 1). The following proposition shows that prior-fitting according to Algorithm 1 achieves the goal of yielding an optimal approximation of the CID from observational data:

**Proposition 1.** *Performing stochastic gradient descent according to Algorithm 1 corresponds to minimizing the expected forward Kullback-Leibler divergence between the conditional interventional distribution $p(y^{in}|\mathbf{x}^{in}, do(t^{in}), \psi)$ and the distribution $q_\theta(y^{in}|do(t^{in}), \mathbf{x}^{in}, \mathcal{D}^{ob})$ parameterized by the model,*

$$\mathbb{E}_{x^{in}, t^{in}, \mathcal{D}^{ob}, \psi}\left[\mathbb{D}_{KL}\left[p(y^{in}|\mathbf{x}^{in}, do(t^{in}), \psi)||q_\theta(y^{in}|do(t^{in}), \mathbf{x}^{in}, \mathcal{D}^{ob})\right]\right]. \tag{3}$$

*Here, the expectation is taken with respect to the data-generating distribution defined in Algorithm 1.*

The proof, given in Appendix A, follows from applying the conditional independences between variables implied by the data-generating process in Algorithm 1.

Let us try to provide some insight about Proposition 1: (i) It does *not* state that we can estimate all causal effects in the traditional sense. To see this, note that the expectation is taken with respect to the synthetic data-generating process. We could even drop the assumption of independent noise terms in our SCMs, to train a model that covers the non-Markovian case, and the proposition would still hold. (ii) Moreover, since our prior over SCMs does *not* necessarily imply identifiability of causal effects, an ideal property of our model would be that $q_\theta(y^{in}|do(t^{in}), \mathbf{x}^{in}, \mathcal{D}^{ob})$ accurately captures the uncertainty in the outcome $y$ arising from the unidentifiability of the causal effect of $do(t^{in})$ on $y^{in}$. Section 4.4 discusses empirical results indicating that Do-PFN is indeed able to do so.

---

[2] We use the *do*-notation in $q_\theta(y^{in}|do(t^{in}), \mathbf{x}^{in}, \mathcal{D}^{ob})$ to indicate that our model approximates the distribution of the outcome $y^{in}$ given an intervention on $t^{in}$. This is formally *not* the result of applying the do-calculus to an observational distribution $q_\theta(y^{in}|t^{in}, \mathbf{x}^{in}, \mathcal{D}^{ob})$.

**Sources of uncertainty** Proposition 1 implies that the learned posterior-predictive distribution $q_\theta(y^{in}|do(t^{in}), \mathbf{x}^{in}, \mathcal{D}^{ob})$ is a forward-KL-optimal approximation of $p(y^{in}|\mathbf{x}^{in}, do(t^{in}), \mathcal{D}^{ob}) = \int p(y^{in}|do(t^{in}), \mathbf{x}^{in}, \psi)\, p(\psi|\mathcal{D}^{ob})d\psi$, in expectation over $p(\mathbf{x}^{in}, t^{in}, \mathcal{D}^{ob})$. The details can be found in Appendix A.1. Furthermore, when defining the observational equivalence relation $\psi_1 \sim \psi_2 : \iff p(y^{in}, \mathbf{x}^{in}, t^{in}|\psi_1) = p(y^{in}, \mathbf{x}^{in}, t^{in}|\psi_2)$, where a selector $r$ maps any SCM $\psi$ onto the representative $r(\psi)$ of its equivalence class $[\psi]$, one can obtain the following:

$$p(y^{in}|\mathbf{x}^{in}, do(t^{in}), \mathcal{D}^{ob}) = \int\int p(y^{in}|\mathbf{x}^{in}, do(t^{in}), \psi)p(\psi|r)p(r|\mathcal{D}^{ob})drd\psi. \tag{4}$$

This factorization allows to distinguish three types of uncertainty that the posterior predictive of Do-PFN captures: First, $p(y^{in}|\mathbf{x}^{in}, do(t^{in}), \psi)$ represents represents *aleatoric uncertainty* about the outcome $y^{in}$ caused by the noise terms in the SCMs. Second, $p(\psi|r)$ is the distribution over observationally equivalent SCMs given equivalence class $[\psi]$, and thus represents *uncertainty due to unidentifiability*. Finally, the distribution $p(r|\mathcal{D}^{ob})$ quantifies the *epistemic uncertainty* about the Markov equivalence class of a ground truth SCM caused by a finite observational dataset $\mathcal{D}^{ob}$.

**Consistency** For an infinite number of observational datapoints, i.e. $|\mathcal{D}^{ob}| \to \infty$, the uncertainty in $p(r|\mathcal{D}^{ob})$ vanishes completely, but the other sources of uncertainty remain. More specifically, when the observational data $\mathcal{D}^{ob}$ comes from an (assumed) ground-truth SCM $\psi_0$, the posterior conditional interventional distribution concentrates up to the Markov-equivalence class $[\psi_0]$ of $\psi_0$:

$$p(y^{in}|\mathbf{x}^{in}, do(t^{in}), \mathcal{D}^{ob}) \to p(y^{in}|\mathbf{x}^{in}, do(t^{in}), [\psi_0]) \quad \text{for } |\mathcal{D}^{ob}| \to \infty. \tag{5}$$

Please refer to Appendix B for a rigorous formulation and a proof of the statement above.

**Architecture and training details** Do-PFN is a transformer with a similar architecture to TabPFN (Hollmann et al., 2025). In order to specialize this architecture for predicting CIDs, we simply add a special indicator to the internal representation of each input dataset to specify that the first column is the treatment and the rest are covariates. Do-PFN has 7.3 million parameters and is trained with Algorithm 1, with details in Appendix C. We primarily evalauate two versions of Do-PFN, `v1` and `v1.1`, which are pretrained for 48 hours and 96 hours respectively on a single RTX 2080.

# 4 Experiments

We thoroughly evaluate Do-PFN's performance and predictive behavior in CID prediction and CATE estimation against a competitive set of causal and tabular machine learning baselines on a combination of synthetic and hybrid synthetic-real-world datasets. First, we introduce our synthetic case studies (Section 4.1), and evaluate Do-PFN against a competitive set of baselines in CID prediction (Section 4.2) as well as CATE (Section 4.3) and ATE estimation. Next, we perform several ablation studies (Section 4.4) across datasets within our prior, conduct an out-of-distribution analysis, and assess the calibration of Do-PFN's uncertainty. Finally, we evaluate Do-PFN on our semi-synthetic datasets (Section 4.5), where Do-PFN shows competitive performance with meta-learner, doubly robust, and deep-learning-based CATE estimators on RealCause, and outperforms them in our known graph case-studies. We provide our pre-trained model, pre-training data generating code, and case study datasets at `https://github.com/jr2021/Do-PFN`.

## 4.1 Synthetic Data

**Case studies** We introduce several causal case studies that pose unique challenges for causal effect estimation, requiring adjustment based on the satisfaction of front-door and back-door criteria (Figure 2). We therefore evaluate Do-PFN's ability to automatically perform this adjustment given only observational data. Please refer to Appendix D.1 for more details.

**Data generation** For each case study visualized in Figure 2, we independently sample 100 datasets with the corresponding graph structure, varying the SCM parameters as described in Appendix C. We also vary the number of samples, standard deviation of noise terms, as well as edge weights and non-linearities. The structural equations for our case studies, as well as details regarding how

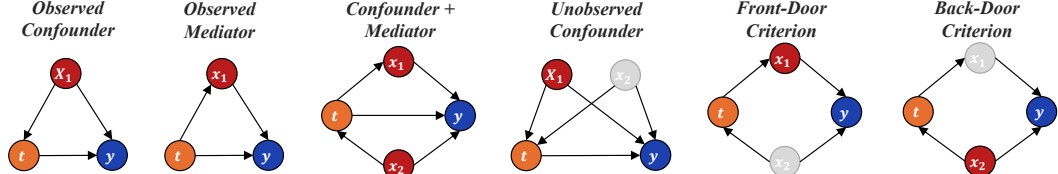

Figure 2: **Case studies**: Visualization of the graph structures of our six causal case studies, requiring Do-PFN to automatically perform adjustment based on the front-door and back-door criteria. Treatment variables $t$ are visualized in orange, covariates $\mathbf{x}$ in red, and outcomes $y$ in blue. Gray variables represent unobservables, not shown to any of the methods yet influencing the generated data.

SCM parameters are sampled, are provided in Appendix D.1 and Appendix Table 1. We additionally generate three case studies not visualized in Figure 2, which ablate over smaller dataset sizes $M_{max} \sim \text{Uniform}([5, 100])$, complex graph structures with number of nodes $K \sim \text{Uniform}([4, 10])$, and finally a "Common Effect" case study which we show to be easily solved even by standard regression models (Appendix Figure 16).

### 4.2 Predicting conditional interventional distributions

First, we evaluate Do-PFN against a competitive set of baselines for the task of predicting the CID $p(y|do(t), \mathbf{x})$. In Figure 3 (first row), we visualize box-plots[3] of normalized mean squared error (MSE) across our case studies. For a description of MSE, please see Appendix D.2.

**Effectiveness of pre-training objective** In Figure 3, we first observe that Do-PFN performs significantly better[4] than the following tabular regression models: Random Forest, TabPFN (v2), as well our own regression model pre-trained on our prior to predict observational outcomes (dubbed "Dont-PFN"). This result offers two interesting findings.

First, the substantial difference in performance between Do-PFN and Dont-PFN provides empirical evidence that our pre-training (Algorithm 1) approximates something fundamentally different from just a standard posterior predictive distribution of observational outcomes, which in turn allows Do-PFN to precisely estimate causal effects.

Second, Do-PFN effectively handles the fact that samples in the *context* $\mathcal{D}^{ob}$ and *query* $\mathcal{D}^{in}$ sets come from systemically different data-generating processes, namely the original SCM and the intervened-upon SCM (in which the incoming edges to the treatment variable are removed). This causes a mismatch in the joint distributions $p(t^{ob}, y^{ob})$ and $p(t^{in}, y^{in})$, precisely when there is a directed edge between treatments $t$ and covariates $\mathbf{x}$. When the treatment and covariates are sampled independently, this mismatch disappears. This is empirically validated by the similar performance of Do-PFN and tabular regression models on the "Common-Effect" case study (Appendix Figure 16). Across other case studies, we observe that TabPFN (v2) and Do-PFN exhibit similar performance when base rate treatment effects are small (Appendix Figure 9).

**Adjustment without graph knowledge** Next, we show that Do-PFN can correctly perform the appropriate form of adjustment without explicit knowledge of the causal graph. When comparing Do-PFN to our "gold standard" baselines DoWhy (Int.) and DoWhy (Cntf.) (Appendix Figure 12), we observe that Do-PFN performs competitively with DoWhy (Int.) in CID prediction. Further, when comparing different variants of Do-PFN in Appendix Figure 15, we observe that even Do-PFN-Short, our shortest pre-trained model, performs competitively with Do-PFN-Graph, a version of Do-PFN which is pre-trained on solely the ground truth graph structure for each case study. These results together exhibit Do-PFN's ability to correctly perform adjustment as if applying the front-door and back-door criteria, while having no knowledge of the graph structure.

---

[3]Our bar-plots visualize median values the 95% confidence interval.
[4]Please refer to the critical difference plots in Appendix E.6. Significance is assessed using a post-hoc Nemenyi test implemented in the Autorank package (Herbold, 2020).

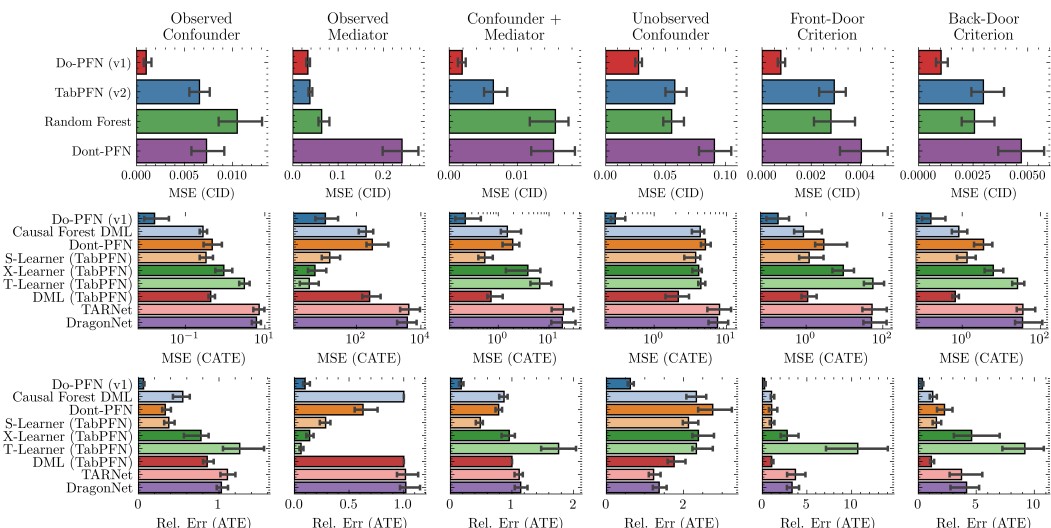

Figure 3: **Results on synthetic data**: Performance of Do-PFN in estimating conditional interventional distributions (CIDs, first row), conditional average treatment effects (CATEs, second row), and average treatment effects (ATEs, third row). Do-PFN provides strong performance across tasks.

## 4.3 Estimating conditional average treatment effects

We now evaluate Do-PFN's ability in CATE estimation, by calculating

$$\hat{\tau}(\mathbf{x}^{in}) = \mathbb{E}_{y^{in} \sim q_\theta(y^{in}|do(1),\mathbf{x}^{in},\mathcal{D}^{ob})}[y^{in}] - \mathbb{E}_{y^{in} \sim q_\theta(y^{in}|do(0),\mathbf{x}^{in},\mathcal{D}^{ob})}[y^{in}]. \tag{6}$$

**Robustness to causal assumptions** In estimating CATE values in settings where the ground truth is known, we observe that Do-PFN significantly outperforms meta-learners (Künzel et al., 2019), a causal forest double machine learning (DML) approach (Wager and Athey, 2018; Chernozhukov et al., 2018), as well as deep-learning-based methods DragonNet and TARNet (Curth et al., 2021), even on our relatively simple case studies (Figure 3, second row). We observe a similar trend in terms of ATE estimation (Figure 3, third row). In Appendix Figure 11, we show that the performance of our CATE baselines depends on the satisfaction of unconfoundedness, which causes our baselines to deviate in performance when this assumption is not met. Do-PFN, on the other hand, remains more consistent in either case, further supporting our previous observation that Do-PFN is able to perform the appropriate adjustment in identifiable scenarios.

**Robustness to unobserved variables** We also observe that in the ground-truth CATE estimation setting, Do-PFN performs closer to the gold standard DoWhy (Cntf.) than previously in the more challenging CID prediction (Appendix Figure 13). Note that Do-PFN is especially competitive on the "Front-Door" and "Back-Door" case studies, where none of the models are given access to the unobserved variable; hence DoWhy loses the fundamental advantage that it had in other settings. This improved performance in the CATE estimation task is further explained in Appendix Figure 14, in which we show via a bias-decomposition that Do-PFN slightly over-predicts the CID in such a way that cancels out when predicting the CATE.

## 4.4 Ablation studies

Furthermore, we perform several ablation studies to elucidate Do-PFN's predictive behavior across datasets with various different causal and statistical characteristics.

**Dataset size and complexity** In an evaluation of MSE in CATE estimation across datasets with a varying number of samples drawn such that $M_{max} \sim \text{Uniform}([5, 2000])$, we observe that Do-PFN performs competitively with DoWhy (Cntf.) and its performance continues to improve and becomes more consistent as dataset size grows (Appendix Figure 9. We also find that Do-PFN performs competitively to DoWhy (Cntf.) across graph complexities, with slightly larger improvements for

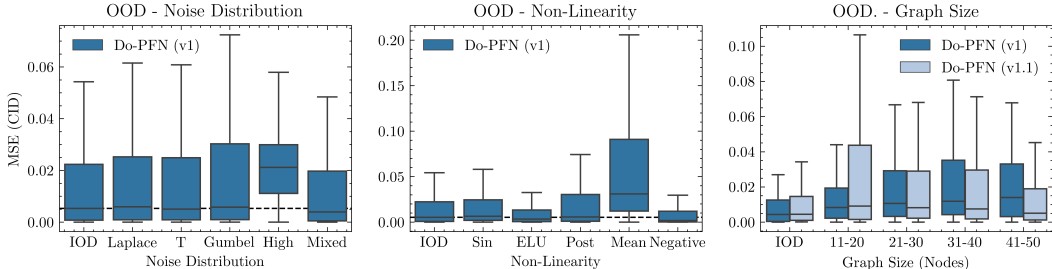

Figure 4: **Out-of-distribution**: Analysis of Do-PFN's performance on 500 in-distribution datasets (IOD) compared to various OOD settings. Do-PFN is robust to different noise distributions (left) and various forms of functional non-linearity (middle). Do-PFN's (v1)'s performance deteriorates on larger graph sizes (right), which is recovered by Do-PFN (v1.1) via larger-scale pre-training.

more complex graphs. Furthermore, we find that Do-PFNs can effectively use additional data points to alleviate increasing levels of noise (Appendix E.2).

**Base rate treatment effect** We also show in Figure 9 in Appendix E.2 that Do-PFN remains consistent in MSE across different base rate levels of the ATE. This result shows that Do-PFN is robust to different magnitudes of base rate ATE, which is also beneficial in cases of problem misspecification, for example when a specified treatment does not influence an outcome.

**Uncertainty calibration** An analysis of prediction interval coverage probability (PICP) curves (Appendix E.5) confirms that Do-PFN is overall well calibrated. In the unidentifiable "Unobserved Confounder" case study, the model's uncertainty increases (Appendix Figure 17), as expected due to unidentifiability (see Section 3). Meanwhile, the PICP curve confirms that Do-PFN learns to exactly capture this increase in uncertainty. We also observe that Do-PFN's is slightly under-confident for theoretically identifiable case studies. Reasons for this include the discretization error resulting from using a bar-distribution to parameterize the posterior and a training loss that is not perfectly minimized. Since Do-PFN minimizes the forward KL-divergence to the ground-truth posterior, this leads to underdispersion Murphy (2023); McNamara et al. (2024).

**Out-of-distribution analysis** We find that Do-PFN's performance is robust to different noise distributions, including noise from Laplace-, T- and Gumbel-distribution, as well as a mixture of all the aforementioned distributions; while using a two-times larger variance of the Gaussian noise drastically hurts performance (leftmost plot in Figure 4). The performance is also relatively robust to using different non-activation functions and to using post-non-linearities (instead of adding the noise after applying a nonlinearity). However, testing on the pointwise average of the functions used during training does lead to a notable decrease in performance. Furthermore, we sample 500 synthetic datasets from our prior with graphs of up to 50 nodes. For larger graphs out of its prior-distribution, we observe a significant reduction in Do-PFN-v1's performance. Do-PFN (v1.1), however, which was trained on graph sizes up to 60 nodes, achieves significantly better performance on large graphs with 21-50 nodes.

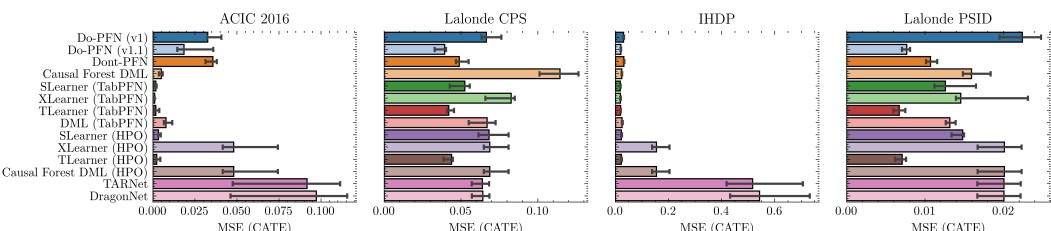

Figure 5: **Results on RealCause**: Performance of Do-PFN and our causal baselines in conditional average treatment effect (CATE) estimation on the RealCause benchmark. Do-PFN provides competitive performance in these semi-synthetic, unconfounded settings.

### 4.5 Hybrid synthetic-real-world data

**RealCause:** In order to evaluate on datasets from the RealCause benchmark (Neal et al., 2020), we pre-train a version of Do-PFN (which we call v1.1) on larger graphs with up to 60 nodes. We observe that Do-PFN-v1.1's extended pre-training leads to strong performance among competitive baselines on three out of four RealCause datasets. These datasets are simulated to perfectly satisfy the unconfoundedness assumption, which inherently advantages the baselines over Do-PFN. Nonetheless, Do-PFN remains competitive with this strong set of CATE estimators, demonstrating that its synthetic pre-training transfers effectively to complex, real-world data. Furthermore, Do-PFN achieves a Pareto-optimal prediction speed and performance compared to all baselines (Appendix E.1).

**Known graph:** Furthermore, we evaluate on the Amazon Sales and Law School Admissions datasets where the causal graph is known (refer to Appendix D.4.2 for more details). We also evaluate against six highly competitive baselines detailed in Appendix D.3. We find that Do-PFN (v1) achieves the strongest performance on Law School and is not significantly worse than the best method on Amazon Sales, a dataset with a complex mediation structure.

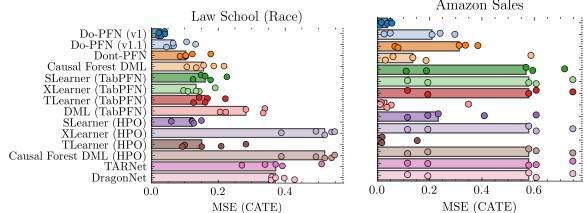

Figure 6: **Results on known graph scenarios**: Performance of Do-PFN and our causal baselines in CATE estimation on the Law School Admissions and Amazon Sales datasets. Do-PFN outperforms other methods in mediated scenarios.

## 5 Discussion

We introduced Do-PFN, a pre-trained transformer leveraging ICL to learn to predict interventional outcomes from observational data. Our empirical results on carefully controlled synthetic setups suggest that Do-PFN performs better in different causal scenarios, and is more robust to the violation of commonly made assumptions, such as unconfoundedness, than a strong set of tabular and causal machine learning baselines. We also perform numerous ablation studies within our prior-distribution to elucidate Do-PFN's predictive behavior, and assess the calibration of our uncertainty estimates—showing both empirically and theoretically that Do-PFN effectively captures the uncertainty that arises from unidentifiability.

Moving outside of our prior distribution, we perform an OOD analysis which shows that Do-PFN is robust to many forms of noise and non-linearities not seen explicitly during pre-training. Our evaluations on two semi-synthetic benchmarks show Do-PFN's competitive performance when unconfoundedness is satisfied, and especially strong performance in mediated scenarios when this assumption is not met. Nevertheless, we need to discuss a range of limitations:

First, Do-PFN's generalization capability critically depends on its synthetic prior over SCMs, $p(\psi)$, adequately capturing real-world causal complexity. As our current validation is based on synthetic data, Do-PFN's robustness to prior-reality mismatches and its performance on diverse real-world datasets require further systematic exploration, alongside principled methods for prior design and validation. What makes us optimistic are empirical findings providing strong evidence that synthetic prior fitting can lead to strong real-world performance (Hollmann et al., 2025; Hoo et al., 2024; Reuter et al., 2025), as well as first theoretical results (Nagler, 2023).

Second, Do-PFN's amortized inference approach offers efficiency but has different theoretical underpinnings compared to traditional, non-amortized estimators under known causal structures. The statistical theory concerning guaranties for such amortized models is still developing.

Third, many configurations and types of causal challenges were necessarily not addressed in this initial work. These include, for instance, a broader array of intervention types beyond binary interventions and counterfactuals, non-i.i.d. input data, and various observational data characteristics or modalities not explicitly part of our current experimental scope. Incorporating them into our data-generating prior can give us a completely new handle on some of these hard problems.

To conclude, the key contribution of Do-PFN is a novel methodology for causal effect estimation. We are optimistic about its prospects to become part of the standard ML toolkit, thus helping to give causal effect estimation the broad accessibility that its real-world relevance deserves.

## Acknowledgements

We acknowledge funding by the European Union via ERC Consolidator Grant DeepLearning 2.0, grant no. 101045765 and via the Horizon Europe research and innovation programme under grant agreement no. 101214398 (ELLIOT). Funded by the European Union. Views and opinions expressed are however those of the author(s) only and do not necessarily reflect those of the European Union, the European Commission, or the European Research Council. Neither the European Union nor the European Commission nor the European Research Council can be held responsible for them. Frank Hutter acknowledges the financial support of the Hector Foundation. This research was also funded by the Deutsche Forschungsgemeinschaft (DFG, German Research Foundation) under grant number 539134284, through EFRE (FEIH_2698644) and the state of Baden-Württemberg.

 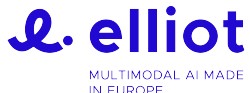 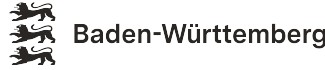

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

# A Proof of Proposition 1

The risk for a single interventional data point when using the NLL loss, as in Algorithm 1 takes the following form:

$$\mathcal{R}_\theta = \int \int \int \int -\log(q_\theta(y^{in}|do(t^{in}), \mathbf{x}^{in}, \mathcal{D}^{ob}))p(\mathcal{D}^{ob}, t^{in}, y^{in}, \mathbf{x}^{in})d\mathcal{D}^{ob}dt^{in}dy^{in}d\mathbf{x}^{in} \quad (7)$$

Let's consider $p(\mathcal{D}^{ob}, t^{in}, y^{in}, \mathbf{x}^{in})$. Then we can obtain by first marginalizing out the distribution $p(\psi)$ of Structural Causal Models (SCMs) and, second, utilizing the factorization of the joint distribution implied by the data generating process in Algorithm 1:

$$p(\mathcal{D}^{ob}, t^{in}, y^{in}, \mathbf{x}^{in}) = \int p(\mathcal{D}^{ob}, t^{in}, y^{in}, \mathbf{x}^{in}, \psi)d\psi =$$
$$\int p(y^{in}, \mathbf{x}^{in}|do(t^{in}), \psi)p(t^{in}|\mathcal{D}^{ob})p(\mathcal{D}^{ob}|\psi)p(\psi)d\psi \quad (8)$$

Now, we can use that

$$p(y^{in}, \mathbf{x}^{in}|do(t^{in}), \psi) = p(y^{in}|\mathbf{x}^{in}, do(t^{in}), \psi)p(\mathbf{x}^{in}|do(t^{in}), \psi).$$

Further:

$$p(\mathbf{x}^{in}|do(t^{in}), \psi)p(t^{in}|\mathcal{D}^{ob})p(\mathcal{D}^{ob}|\psi)p(\psi) = p(\mathcal{D}^{ob}, t^{in}, \mathbf{x}^{in}, \psi). \quad (9)$$

This implies

$$p(\mathcal{D}^{ob}, t^{in}, y^{in}, \mathbf{x}^{in}) = \int p(y^{in}|\mathbf{x}^{in}, do(t^{in}), \psi)p(\mathcal{D}^{ob}, t^{in}, \mathbf{x}^{in}, \psi)d\psi \quad (10)$$

Plugging this into equation 7 followed by using that the cross entropy between two distributions $p$ and $q$ is equal to the Kullback-Leibler divergence between $p$ and $q$ plus the entropy of $p$, formally $H(p, q) = H(p) + \mathbb{D}_{KL}(p||q)$, a fact used by Müller et al. (2022) and Barber and Agakov (2004) in analogous scenarios, yields:

$$\mathcal{R}_\theta = \int \int \int \int \int -\log(q_\theta(y^{in}|do(t^{in}), \mathbf{x}^{in}, \mathcal{D}^{ob}))$$
$$p(y^{in}|\mathbf{x}^{in}, do(t^{in}), \psi)p(\mathcal{D}^{ob}, t^{in}, \mathbf{x}^{in}, \psi)d\mathcal{D}^{ob}dt^{in}dy^{in}d\mathbf{x}^{in}d\psi$$
$$= \int \int \int \int \int \mathbb{D}_{KL}\left[p(y^{in}|\mathbf{x}^{in}, do(t^{in}), \psi)||q_\theta(y^{in}|do(t^{in}), \mathbf{x}^{in}, \mathcal{D}^{ob})\right]$$
$$p(\mathcal{D}^{ob}, t^{in}, \mathbf{x}^{in}, \psi)d\mathcal{D}^{ob}dt^{in}d\mathbf{x}^{in}d\psi + C \quad (11)$$

This implies that minimizing $\mathcal{R}_\theta$ results in a (forward) Kullback-Leibler optimal approximation of $p(y^{in}|do(t^{in}), \psi, \mathbf{x}^s)$ with the model $q_\theta(y^{in}|do(t^{in}), \mathbf{x}^s, \mathcal{D}^{ob})$ **in expectation** over the data simulated from $p(\psi, \mathcal{D}^{ob}, t^{in}, \mathbf{x}^{in})$.

Please note that analogous to PFNs, the optimality only holds when the expectation is taken with respect to the synthetic data-generating process. However, theoretical results by Nagler (2023) and a plethora of empirical findings regarding the transferability of PFNs to real-world scenarios, as well as related approaches (Hollmann et al., 2025; Hoo et al., 2024; Reuter et al., 2025), provide evidence that synthetic prior fitting can lead to strong real-world performance.

## A.1 Alternative version of proposition 1

Furthermore, when using

$$p(y^{in}|do(t^{in}), \mathbf{x}^{in}, \mathcal{D}^{ob}) := \int p(y^{in}|do(t^{in}), \mathbf{x}^{in}, \psi) \, p(\psi|\mathcal{D}^{ob}) d\psi, \qquad (12)$$

as for instance in Madigan and Raftery (1994), we can rewrite:

$$\mathcal{R}_\theta = \int \int \int \int \int -\log(q_\theta(y^{in}|do(t^{in}), \mathbf{x}^{in}, \mathcal{D}^{ob}))$$
$$p(y^{in}|\mathbf{x}^{in}, do(t^{in}), \psi) p(\mathcal{D}^{ob}, t^{in}, \mathbf{x}^{in}, \psi) d\mathcal{D}^{ob} dt^{in} dy^{in} d\mathbf{x}^{in} d\psi =$$
$$\int \int \int \int -\log(q_\theta(y^{in}|do(t^{in}), \mathbf{x}^{in}, \mathcal{D}^{ob}))$$
$$\int p(y^{in}|\mathbf{x}^{in}, do(t^{in}), \psi) p(\psi|\mathcal{D}^{ob}, \mathbf{x}^{in}) d\psi \, p(\mathcal{D}^{ob}, t^{in}, \mathbf{x}^{in}) d\mathcal{D}^{ob} dt^{in} dy^{in} d\mathbf{x}^{in} =$$
$$\int \int \int \int -\log(q_\theta(y^{in}|do(t^{in}), \mathbf{x}^{in}, \mathcal{D}^{ob}))$$
$$p(y^{in}|do(t^{in}), \mathbf{x}^{in}, \mathcal{D}^{ob}) \, p(\mathcal{D}^{ob}, t^{in}, \mathbf{x}^{in}) d\mathcal{D}^{ob} dt^{in} dy^{in} d\mathbf{x}^{in}, \quad (13)$$

which, using the same arguments as before implies an alternative version of Proposition 1:

**Proposition 2.** *Performing stochastic gradient descent according to Algorithm 1 corresponds to minimizing the expected forward Kullback-Leibler divergence between $p(y^{in}|do(t^{in}), \mathbf{x}^{in}, \mathcal{D}^{ob})$ and the distribution $q_\theta(y^{in}|do(t^{in}), \mathbf{x}^{in}, \mathcal{D}^{ob})$ parameterized by the model,*

$$\mathbb{E}_{x^{in}, t^{in}, \mathcal{D}^{ob}} \big[ \mathbb{D}_{KL} \big[ p(y^{in}|do(t^{in}), \mathbf{x}^{in}, \mathcal{D}^{ob}) || q_\theta(y^{in}|do(t^{in}), \mathbf{x}^{in}, \mathcal{D}^{ob}) \big] \big]. \qquad (14)$$

Note that, in comparison to Proposition 1, the expectation is **no longer** taken over the SCMs.

## B   On the consistency of Do-PFN

In the following section, we will show that the posterior predictive interventional distribution $P(Y^{in} | do(T^{in}), \mathbf{X}^{in}, \mathcal{D}^{ob})$ of Do-PFN is consistent up to observationally equivalent SCMs. Assuming the observational data $\mathcal{D}^{ob}$ were generated by an SCM $\Psi_0$, but could also have been generated by the set $[\Psi_0]$, we have (informally) that when the number of observational data points goes to infinity $|\mathcal{D}^{ob}| \to \infty$.

$$P(Y^{in} | do(t^{in}), \mathbf{X}^{in}, \mathcal{D}^{ob}) \to P(Y^{in} | do(T^{in}), \mathbf{X}^{in}, [\Psi_0]). \qquad (15)$$

This means that we asymptotically recover the posterior $P(Y^{in} | do(T^{in}), \mathbf{X}^{in}, [\Psi_0])$ under the set of all SCMs $[\Psi_0]$ that yield the same observational distribution $P(\mathcal{D}^{ob}|\Psi_0) = (\mathcal{D}^{ob}|\Psi_0')$ for all $\Psi_0' \in [\Psi_0]$.

To prove this fact, we use Doob's theorem (Doob, 1949; Miller, 2018) and ideas that also appear in a proof by (Balazadeh et al., 2025) which pertains to the consistency of posterior expectations.

### B.1   Mathematical framework

In the following, we depart from the previous convention of exclusively working with probability densities and use capital letters to denote random variables with capital $P$ denoting a probability distribution with, if it exists, density $p$ wrt. the Lebesgue measure or counting measure (if $p$ is a probability mass function) respectively.

Further, let $(\mathcal{Y}, \mathcal{B})$ be the outcome space for $Y^{in}$. Let $\Psi$ be the random SCM under a prior $P(\Psi)$ over SCMs, and let

$$D_n = \big[ (Y_j^{ob}, T_j^{ob}, \mathbf{X}_j^{in}) \big]_{j=1}^n$$

be the first $n$ observational i.i.d. samples from the observational distribution $P(Y^{ob}, T^{ob}, \boldsymbol{X}^{in} | \Psi = \Psi_0)$ under a true SCM $\psi_0$. We assume throughout that all random variables take values in complete separable metric spaces equipped with the induced Borel $\sigma$-algebras. Note that this requirement is satisfied for the simple case of real-valued (multivariate) random variables which represent the only relevant case for Do-PFN.

Define observational equivalence $\sim$ on the domain of $\Psi$ by

$$\psi \sim \phi \;:\Longleftrightarrow\; P\big(Y^{ob}, T^{ob}, \boldsymbol{X}^{in} \mid \Psi = \psi\big) = P\big(Y^{ob}, T^{ob}, \boldsymbol{X}^{in} \mid \Psi = \phi\big).$$

Choose a measurable selector $r : \Psi \to \mathcal{R} \subseteq \Psi$ so that $r(\psi)$ is the unique representative of $[\psi]$, and set

$$R = r(\Psi) \quad r_0 = r(\psi_0).$$

Please note that we use this set of representatives, and the distribution $P(R) = r\sharp P(\Psi)$, which is the pushforward of $P(\Psi)$ under $r$, to avoid working with set-valued random variables.

We write $P\big(Y^{in} \mid do(T^{in}), \boldsymbol{X}^{in}, [\psi_0]\big)$ to denote $P\big(Y^{in} \mid do(T^{in}), \boldsymbol{X}^{in}, R = r_0\big)$.

Finally, define for each $B \in \mathcal{B}$:

$$\mu_n(B) = P\big(Y^{in} \in B \mid do(T^{in}), \boldsymbol{X}^{in}, D_n\big), \qquad \mu_\infty(B) = P\big(Y^{in} \in B \mid do(T^{in}), \boldsymbol{X}^{in}, R = r_0\big).$$

**Proposition 3** (Consistency of Do-PFN). *Assume observational data $D_n$ drawn i.i.d. drawn from an SCM $\Psi_0$, i.e. $D_n \overset{\text{iid}}{\sim} P\big(Y^{ob}, T^{ob}, \boldsymbol{X}^{ob} \mid \Psi = \Psi_0\big)$. Then, for sample size $n \to \infty$,*

$$P\big(Y^{in} \mid do(T^{in}), \boldsymbol{X}^{in}, D_n\big) \xrightarrow{w} P\big(Y^{in} \mid do(T^{in}), \boldsymbol{X}^{in}, \tilde{\psi}_0\big),$$

*where $\xrightarrow{w}$ denotes weak convergence of measures.*

*Proof.* Fix any Borel $B \subseteq \mathcal{Y}$ and define

$$M_n(B) = P\big(Y^{in} \in B \mid do(T^{in}), \boldsymbol{X}^{in}, D_n\big) = \mathbb{E}\big[\mathbf{1}_{\{Y^{in} \in B\}} \mid D_n, do(T^{in}), \boldsymbol{X}^{in}\big].$$

**1. First tower step.** By the law of total expectation,

$$M_n(B) = \mathbb{E}\Big[\mathbb{E}\big[\mathbf{1}_{\{Y^{in} \in B\}} \mid D_n, do(T^{in}), \boldsymbol{X}^{in}, \Psi\big] \,\Big|\, D_n, do(T^{in}), \boldsymbol{X}^{in}\Big].$$

Since knowing $\Psi$ renders the past data $D_n$ irrelevant for the conditional interventional distribution,

$$\mathbb{E}\big[\mathbf{1}_{\{Y^{in} \in B\}} \mid D_n, do(T^{in}), \boldsymbol{X}^{in}, \Psi\big] = \mathbb{E}\big[\mathbf{1}_{\{Y^{in} \in B\}} \mid do(T^{in}), \boldsymbol{X}^{in}, \Psi\big].$$

Furthermore, since $P(\Psi | D_n, \boldsymbol{X}^{in}, do(T^{in})) = P(\Psi | D_n)$ due to the conditional independencies in algorithm 1:

$$\mathbb{E}\Big[\mathbb{E}\big[\mathbf{1}_{\{Y^{in} \in B\}} \mid D_n, do(T^{in}), \boldsymbol{X}^{in}, \Psi\big] \,\Big|\, D_n, do(T^{in}), \boldsymbol{X}^{in}\Big] = \mathbb{E}\Big[\mathbb{E}\big[\mathbf{1}_{\{Y^{in} \in B\}} \mid D_n, do(T^{in}), \boldsymbol{X}^{in}, \Psi\big] \,\Big|\, D_n\Big]$$

When assuming that densities exist, the steps above just represent the fact that $p(y^{in}|do(t^{in}), \mathbf{x}^{in}, \mathcal{D}^{ob}) = \int p(y^{in}|do(t^{in}), \mathbf{x}^{in}, \psi)\, p(\psi|\mathcal{D}^{ob}) d\psi$.

**2. Second tower step.** Conditioning next on the representative $R = r(\Psi)$,

$$M_n(B) = \mathbb{E}\Big[\mathbb{E}\big[\mathbb{E}\big[\mathbf{1}_{\{Y^{in} \in B\}} \mid do(T^{in}), \boldsymbol{X}^{in}, \Psi\big] \,\Big|\, D_n\big] \,\Big|\, R\Big] = \mathbb{E}\big[g_B(R) \mid D_n\big],$$

where

$$g_B(r) := \mathbb{E}\big[\mathbf{1}_{\{Y^{in} \in B\}} \mid do(T^{in}), \boldsymbol{X}^{in}, R = r\big] = P\big(Y^{in} \in B \mid do(T^{in}), \boldsymbol{X}^{in}, R = r\big).$$

In terms of densities, this can be seen via:

$$p(\psi \mid \mathcal{D}_n) = \int p(r \mid \mathcal{D}_n)\, p(\psi \mid r, \mathcal{D}_n)\, dr,$$

and noting by definition of observational equivalence $\sim$ that $p(\psi \mid r, \mathcal{D}_n^{ob}) = p(\psi \mid r)$, we have $p(\psi \mid \mathcal{D}_n) = \int p(r \mid \mathcal{D}_n^{ob}) \, p(\psi \mid r) \, dr$. This can be plugged into $\int p(y^{in} | do(t^{in}), \mathbf{x}^{in}, \psi) \, p(\psi | \mathcal{D}^{ob}) d\psi$ to get:

$$p(y^{in} | do(t^{in}), \mathbf{x}^{in}, \mathcal{D}^{ob}) = \int \int p(y^{in} | do(t^{in}), \mathbf{x}^{in}, \psi) \; p(\psi \mid r) \; d\psi \; p(r \mid \mathcal{D}_n^{ob}) \, dr$$

**3. Martingale convergence.** By construction, we have that $P\big(Y^{ob}, T^{ob}, \boldsymbol{X}^{in} \mid \Psi = r\big) = P\big(Y^{ob}, T^{ob}, \boldsymbol{X}^{in} \mid \Psi = r'\big)$. whenever $r \neq r'$. Furthermore, since $r \to P\big(Y^{in} \in B \mid do(T^{in}), \boldsymbol{X}^{in}, R = r\big)$ is measurable since it can be written in terms of conditional expecations, we can apply Doob's martingale convergence theorem (Miller, 2018) which yields:

$$M_n(B) \xrightarrow{\text{a.s.}} g_B\big(r_0\big) = g_B(r(\psi_0)),$$

i.e., $\mu_n(B) \to \mu_\infty(B)$ a.s. for every measurable set $B$. This implies $\mu_n \to \mu_\infty$ weakly. $\qquad \square$

### B.2 Comments on the consistency theorem 3

Previously, we have shown that when observational data $\mathcal{D}_n^{ob}$ are generated by an SCM $\psi_0$

$$P\big(y^{in} \mid do(t^{in}), \mathbf{x}^{in}, \mathcal{D}_n^{ob}\big) \to P\big(y^{in} \mid do(t^{in}), \mathbf{x}^{in}, [\psi_0]\big). \tag{16}$$

Note that $P\big(y^{in} \mid do(t^{in}), \mathbf{x}^{in}, [\psi_0]\big)$ has the following density:

$$p\big(y^{in} \mid do(t^{in}), \mathbf{x}^{in}, [\psi_0]\big) = \int p\big(y^{in} \mid do(t^{in}), \mathbf{x}^{in}, \psi\big) p(\psi|r) \, d\psi, \tag{17}$$

where $r$ denotes the representative of the observational equivalence class of $\psi_0$. Further, $p(\psi|r) = p(\psi|[\psi_0])$ specifies how likely the specific SCM $\psi$ is given that the SCM is in the observational equivalence class $[\psi_0]$.

Note that when $[\psi_0] = \{\psi_0\}$, i.e. there exists exactly one SCM $\psi_0$ that could have generated the observational data $\mathcal{D}_n^{ob}$, meaning that $\psi_0$ is identifiable from $P(\mathcal{D}_n^{ob})$, we get that $P(\psi|[\psi_0]) = \delta_{\psi_0}$ and thus recover the conditional international distribution under the unique $\psi_0$ for infinite observational data:

$$P\big(y^{in} \mid do(t^{in}), \mathbf{x}^{in}, \mathcal{D}_n^{ob}\big) \to P\big(y^{in} \mid do(t^{in}), \mathbf{x}^{in}, \psi_0\big). \tag{18}$$

## C  Details on the prior-fitting procedure

In this section, we provide the details of the data-generating process in Algorithm 1 that represents our modeling assumptions. From the perspective of PFNs, this data-generating process represents Do-PFN's "prior". Concretely, our prior-fitting procedure involves the following key steps:

**Sampling the SCM:**   First, for every iteration $i = 1, 2, \ldots, N$, an SCM $\psi_i$ is sampled. This is achieved by first sampling a DAG via topological sorting of vertices (Manber, 1989). For each node $k$ in the graph, we uniformly at random sample the nonlinearity $\gamma$ to be one of the following functions: the quadratic function $x \mapsto x^2$, $x \mapsto \text{ReLU}(x)$, and $x \mapsto \tanh(x)$. We define the mechanisms in the SCM to take the form of an additive noise model (ANM) $f_k(z_{\text{PA}(k)}, \epsilon_k) = \gamma(\sum_{l \in \text{PA}(k)} w_l z_l) + \epsilon_k$. The weights of the SCM are sampled using a Kaiming initialization $w_l \sim \text{Uniform}(-\frac{1}{\sqrt{|\text{PA}(k)|}}, \frac{1}{\sqrt{|\text{PA}(k)|}})$ for $l = 1, 2, \ldots, |\text{PA}(k)|$, where $|\text{PA}(k)|$ denotes the number of parents of node $k$.

**Sampling observational data:**   Next, observational data is sampled according to the SCM $\psi_i$. More specifically, a dataset $\mathcal{D}_i^{ob}$ is filled with $M_{ob}$ data points, where the number of data points is drawn uniformly between $M_{min} = 10$ and $M_{max} = 2,200$. Each element in $\mathcal{D}_i^{ob}$ is generated by first sampling a noise vector $\epsilon_j \sim p(\epsilon)$ which is passed through the SCM to generate each element $y_j^{ob}, t_j^{ob}, \mathbf{x}_j^{ob}$.

**Sampling interventional data:** To sample an element in the interventional dataset $\mathcal{D}_i^{in}$, with $M^{in} = M_{max} - M^{ob}$ data points, first, a noise vector $\epsilon_k \sim p(\epsilon)$ is sampled again. Subsequently a covariate-vector $\mathbf{x}_k^{in}$ is sampled from $p(\mathbf{x}|\psi_i, \epsilon_k)$. This ensures that the vector $\mathbf{x}_k^{in}$ characterizes the subject $k$ prior to the intervention. After sampling the value for the treatment $t_k^{in}$, we perform the intervention $do(t_k^{in})$ and sample $y_k^{in}$ from the intervened-upon SCM using the same noise $\epsilon_k$ as before[5].

**Gradient descent** For each iteration $i = 1, 2, \ldots, N$, an observational dataset $\mathcal{D}_i^{ob}$ and an interventional dataset $\mathcal{D}_i^{in}$ are generated. These datasets are utilized to compute the negative log-likelihood under our model $q_\theta$. This loss is calculated with respect to predicting the interventional outcome $y_k^{in}$ based on the value of the intervention $t_k^{in}$, the covariates $\mathbf{x}_k^{in}$, and the observational dataset $\mathcal{D}_i^{ob}$. Subsequently, a gradient step is taken on the negative log-likelihood. In practice, we perform mini-batch stochastic gradient descent using the Adam optimizer (Kingma and Ba, 2014).

# D    Experimental Details

## D.1    Details on the synthetic case studies

In this section we provide the details on all considered case studies from Section 4.1. The standard deviation $\sigma_{exo}$ of the exogenous noise is sampled from $\sigma_{exo} \sim \text{Uniform}([1, 3])$. For the standard deviation of the additive noise terms, we sample $\beta \sim \text{Beta}(1, 5)$, and then set $\sigma_\epsilon = 0.3 \cdot \beta$.

The functions $f_{z_k}$ take the form $f_a(z_k, \epsilon) = \gamma(\sum_{l \in \text{PA}(k)} w_l z_l) + \epsilon$. The weights of the SCM are sampled using a Kaiming initialization $w_l \sim \text{Uniform}(-\frac{1}{\sqrt{|\text{PA}(k)|}}, \frac{1}{\sqrt{|\text{PA}(k)|}})$ for $l = 1, 2, \ldots, |\text{PA}(k)|$, where $|\text{PA}(k)|$ denotes the number of parents of node $k$. The nonlinearities $f_a$ are sampled uniformly at random from the set $\{f_1, f_2, f_3\}$ where $f_1(x) = x^2$, $f_2(x) = \tanh(x)$ and $f_3 = ReLU(x) = \max(0, x)$. Details on the case studies can be found in Table 1.

---

[5]Because the noise is held constant to produce the pre-interventional covariate-vector, $\mathbf{x}_k^{in}$, and interventional outcomes, $y_k^{in}$, this process can also be seen as simulating counterfactuals or single potential outcomes.

| Setting | Equations |
|---------|-----------|
| Observed Confounder | $\epsilon_t, \epsilon_y \sim \mathcal{N}(0, \sigma_\epsilon)$ 
 $x_1 \sim \mathcal{N}(0, \sigma_{exo})$ 
 $t = f_t(x_1, \epsilon_t)$ 
 $y = f_y(x_1, t, \epsilon_y)$ |
| Observed Mediator | $\epsilon_{x_1}, \epsilon_y \sim \mathcal{N}(0, \sigma_\epsilon)$ 
 $t \sim \text{Uniform}(\{0, 1\})$ 
 $x_1 = f_{x_1}(t, \epsilon_{x_1})$ 
 $y = f_y(x_1, t, \epsilon_y)$ |
| Confounder + Mediator | $\epsilon_{x_1}, \epsilon_t, \epsilon_y \sim \mathcal{N}(0, \sigma_\epsilon)$ 
 $x_2 \sim \mathcal{N}(0, \sigma_{exo})$ 
 $t = f_t(x_1, \epsilon_t)$ 
 $x_1 = f_{x_1}(t, \epsilon_{x_1})$ 
 $y = f_y(x_1, x_2, t, \epsilon_y)$ |
| Unobserved Confounder | $\epsilon_t, \epsilon_y \sim \mathcal{N}(0, \sigma_\epsilon)$ 
 $x_1, x_2 \sim \mathcal{N}(0, \sigma_{exo})$ 
 $t = f_t(x_1, x_2, \epsilon_t)$ 
 $x_1 = f_{x_1}(t, \epsilon_{x_1})$ 
 $y = f_y(x_1, x_2, t, \epsilon_y)$ |
| Back-Door Criterion | $\epsilon_{x_1}, \epsilon_t, \epsilon_y \sim \mathcal{N}(0, \sigma_\epsilon)$ 
 $x_2 \sim \mathcal{N}(0, \sigma_{exo})$ 
 $t = f_t(x_1, \epsilon_t)$ 
 $x_1 = f_{x_1}(x_2, \epsilon_{x_1})$ 
 $y = f_y(x_1, x_2, \epsilon_y)$ |
| Front-Door Criterion | $\epsilon_{x_1}, \epsilon_t, \epsilon_y \sim \mathcal{N}(0, \sigma_\epsilon)$ 
 $x_2 \sim \mathcal{N}(0, \sigma_{exo})$ 
 $t = f_t(x_1, \epsilon_t)$ 
 $x_1 = f_{x_1}(x_2, \epsilon_{x_1})$ 
 $y = f_y(x_1, x_2, \epsilon_y)$ |

Table 1: Structural equations for all causal case studies.

## D.2 Evaluation metric

We evaluate our results in terms of normalized mean squared error (MSE), as it allows results to be compared across datasets. We define MSE below:

$$\text{MSE}(\mathbf{y}, \hat{\mathbf{y}}) = \frac{1}{n} \sum_{i=1}^{n} \left[ \frac{y_i - \hat{y}_i}{\max(\mathbf{y}) - \min(\mathbf{y})} \right]^2 \tag{19}$$

## D.3 Description of baselines

### D.3.1 Conditional interventional distribution prediction

- **Do-PFN (v1) (ours)**: a TabPFN regression model pre-trained for 48 hours on varying graph structures of up to 10 nodes to approximate the CID $p(y^{do}|\mathbf{x}^{ob}, \mathcal{D}^{ob})$

- **Dont-PFN**: a TabPFN regression model (Hollmann et al., 2025) pre-trained on our prior to approximate the posterior predictive distribution (PPD) $p(y^{ob}|\mathbf{x}^{ob}, \mathcal{D}^{ob})$.

- **Do-PFN-Short**: a TabPFN regression model pre-trained for 20 hours on varying graph structures of up to 5 nodes to approximate the CID $p(y^{do}|\mathbf{x}^{ob}, \mathcal{D}^{ob})$

- **Do-PFN-Graph**: a TabPFN regression model pre-trained for 5 hours to approximate the CID $p(y^{do}|\mathbf{x}^{ob}, \mathcal{D}^{ob})$ on *fixed* graph structures from our case studies.

- **Do-PFN-Mixed**: Do-PFN-Short pre-trained for 24 hours varying whether additive noise terms are sampled from zero-mean Gaussian, Laplacian, Students-T, and Gumbel distributions.

- **DoWhy (Int./Cntf.)**: a SCM $\psi$ fit to observational samples $\mathcal{D}^{ob}$ and the graph structure $\mathcal{G}_\psi$. The constructed SCM is used to predict interventional (Int.) and counterfactual (Cntf.) outcomes. Crucially, TabPFNClassifier and TabPFNRegressor models (Hollmann et al., 2025) are used approximate binary and continuous structural equations.

- **TabPFN (v2)**: A TabPFNRegressor model as proposed in (Hollmann et al., 2025) pretrained to approximate the posterior predictive distribution (PPD) $p(y^{ob}|\mathbf{x}^{ob}, \mathcal{D}^{ob})$.

- **Random Forest**: an ensemble of decision trees Breiman (2001) trained on $\mathcal{D}^{ob}$

### D.3.2 Conditional average treatment effect estimation

- **Do-PFN (v1) (ours)**: Do-PFN (v1) applied to predict the specific quantity

$$\hat{\tau} = \mathbb{E}_{y^{in} \sim q_\theta(y^{in}|do(t^{in}=1), \mathbf{x}^{in}, \mathcal{D}^{ob})}[y^{in}] - \mathbb{E}_{y^{in} \sim q_\theta(y^{in}|do(t^{in}=0), \mathbf{x}^{in}, \mathcal{D}^{ob})}[y^{in}], \tag{20}$$

where the model directly approximates the conditional interventional distribution.

- **Do-PFN (v1.1) (ours)**: a version of Do-PFN pre-trained for 96 hours on varying graph structures of up to 60 nodes applied to predict the CATE as described above.

- **Dont-PFN**: Dont-PFN applied to predict the CATE.

- **S-Learner (TabPFN)**: a meta-learning approach (Künzel et al., 2019) that trains a single TabPFN model jointly on treatment and covariates to predict outcomes, using the treatment as an additional input feature.

- **T-Learner (TabPFN)**: a two-model meta-learning approach (Künzel et al., 2019) that fits separate TabPFN predictors for treated and control populations, estimating CATEs from their prediction difference.

- **X-Learner (TabPFN)**: an extension of the meta-learning framework (Künzel et al., 2019) where TabPFN serves as the base model in a cross-fitting procedure to reduce bias and improve finite-sample efficiency.

- **DML (TabPFN)**: a double machine learning (DML) baseline using TabPFN as the base learner, following (Nie and Wager, 2021), which leverages orthogonalized score functions and flexible meta-learned regressors to estimate treatment effects.

- **S-Learner (HPO)**: a hyperparameter-optimized meta-learning baseline implemented using the EconML library (Battocchi et al., 2019), where a single model jointly learns to predict outcomes from covariates and treatment. HPO is performed using FLAML where we tune the the base-learner for 60 seconds (Wang et al., 2021).

- **T-Learner (HPO)**: a hyperparameter-optimized two-model baseline implemented using EconML, fitting separate outcome models for treated and control units. HPO is performed using FLAML where we tune the each base-learner for 60 seconds (Wang et al., 2021).

- **X-Learner (HPO)**: a hyperparameter-optimized variant of the X-Learner (Künzel et al., 2019), implemented with EconML, which combines treatment and control outcome models through cross-fitting to reduce estimation bias. HPO is performed using FLAML where we tune the the base-learners and the propensity model each for 60 seconds (Wang et al., 2021).

- **TARNet**: the treatment-agnostic representation network (Shalit et al., 2017), which learns shared covariate representations and separate outcome heads for treated and control groups; implemented using the CATENets library (Curth et al., 2021).

- **DragonNet**: a neural network–based CATE estimator (Shi et al., 2019) that augments TARNet with an explicit propensity head and targeted regularization to improve covariate balance; also implemented using CATENets.

- **Causal Forest (DML)**: a double machine learning (DML) approach based on (Wager and Athey, 2018) that combines multiple causal trees to estimate conditional average treatment effects (CATEs). Hyperparameters are tuned using exhaustive search.

- **DoWhy-CATE (Int./Cntf.)**: DoWhy (Int./Cntf.) used as an S-Learner (Künzel et al., 2019) to estimate conditional average treatment effects (CATEs). When DoWhy (Cntf.) is used, noise terms are inferred and held constant across forward passes.

### D.3.3 Software

We use Pytorch (Paszke, 2019) to implement all our experiments. Our implementation of the causal prior is based on the Causal Playground library (Sauter et al., 2024) and the codebase used for TabPFN (Hollmann et al., 2023, 2025). We use Matplotlib (Hunter, 2007), Autorank (Herbold, 2020) and Seaborn (Waskom, 2021) for our plots.

### D.4 Hybrid synthetic-real-world data

### D.4.1 RealCause

The RealCause benchmark (Neal et al., 2020) provides realistic causal inference datasets by fitting generative models to real-world data, ensuring that the simulated data is similar to the real-world data. This approach enables access to ground-truth causal effects while preserving characteristics of the real-world data. Crucially, all observational datasets are simulated in a way that ensures that the unconfoundedness assumptions strictly hold, regardless of whether the actual real-world data supports this.

### D.4.2 Known graph

We furthermore conduct experiments on two real-world datasets with agreed upon causal graphs depicted in Figure 7. Those causal graphs allow us to simulate gold-standard outcomes using the DoWhy library (Sharma and Kiciman, 2020), which facilitates the evaluation of Do-PFN and our baselines. More concretely, we specify the agreed upon causal graph and use DoWhy to estimate the causal mechanisms from the real-world datasets. Subsequently, we simulate interventional data using these mechanisms and use the real-world observational data as input to all models.

**Amazon Sales** The Amazon sales dataset (Blöbaum et al., 2024) contains data on the effect of special shopping events ("Shopping Event?") on the profit made from smartphone sales ("Profit"). It further provides variables with information on the spending on ad campaigns ("Ad Spend"), the price of the device ("Unit Price"), the number of phones sold ("Sold Units"), the number of page view ("Page Views"), the revenue that day ("Revenue") and the operational cost ("Operational Cost").

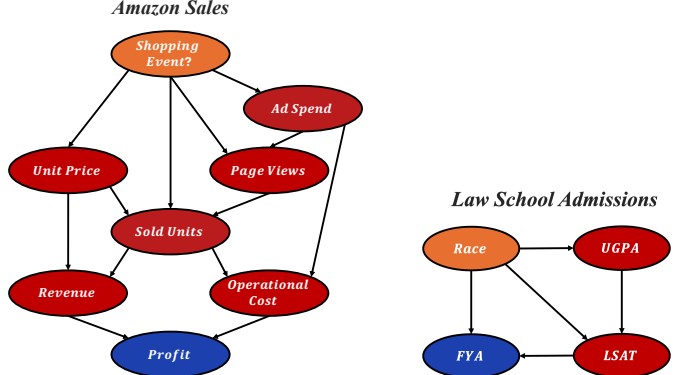

Figure 7: **Real-world case studies**: Widely agreed-upon causal graphs for the Amazon Sales and Law School Admissions datasets.

**Law School Admissions** The law school admissions dataset (Figure 1) was drawn from the 1998 LSAC National Longitudinal Bar Passage Study (Wightman, 1998) and was made popular in the realm of counterfactual fairness due to its appearance in Kusner et al. (2017) , where the variable "Race" was treated as a protected attribute. We note that we do not address the topic of algorithmic fairness, but would like to highlight that the ability of Do-PFN to predict interventional outcomes on demographic information could be a fruitful application in model-bias assessment. We delve into the effect 32 of "Race" [6] on first-year-average ("FYA"), which is mediated by two variables undergraduate grade-point-average ("UGPA") and "LSAT", a law school entrance exam in the United-States.

# E Supplementary results

## E.1 Inference speed

While TabPFN's synthetic pre-training paradigm offers benefits in terms of predictive performance and generalization, it also offers benefits in terms of training (`.fit()`) and inference (`.predict()`), as the cost of pre-training is incurred once up front such that training examples can be passed into the transformer *in-context* and predictions can be obtained in a single forward pass. We demonstrate on the RealCause benchmark, a collection of up to 100 realizations of each of four datasets that range from 10-50 features and up to 10,000 samples, that Do-PFN achieves Pareto-Optimal prediction speed and performance across all four dataset groups.

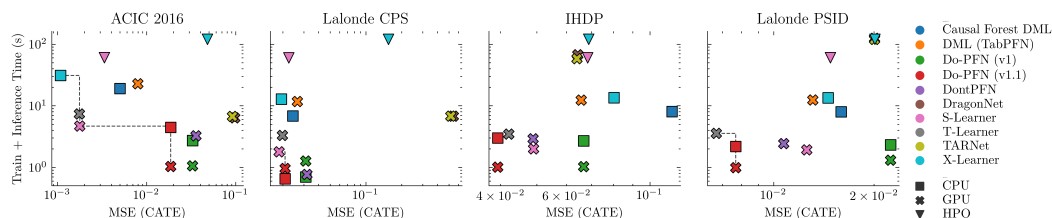

Figure 8: **Training and inference speed**: Multi-objective scatter plot and Pareto Fronts depicting average inference speed and normalized mean squared error (MSE) of Do-PFN-CATE and our causal baselines in conditional average treatment effect (CATE) estimation on the RealCause benchmark. Do-PFN is on the Pareto Front on all RealCause datasets.

---

[6]We note that Race in the lawschool dataset is typically treated as a binary variable. We very much disagree with this formulation, and acknowledge that the term "ethnicity" better describes this complex social construct.

## E.2 In-distribution analysis

In order to better understand Do-PFN's behavior on different parts of its' prior, we conduct several ablation studies varying the base rate of the average treatment effect, dataset size, as well as graph size and complexity.

**Base rate average treatment effect** First, we evaluate Do-PFN's performance on datasets with various base rates of the average treatment effect (ATE). We observe in Figure 9 (left) that while Do-PFN and TabPFN (v2) perform similarly on datasets with a small ATE, TabPFN's performance diverges as the base ATE grows. This divergence can be explained by either Do-PFN's pre-training objective or its ability to handle the implicit distribution shift between the context and query set created when predicting interventional outcomes. We do note, however, that Do-PFN's performance also seems to diverge in the largest ATE group. This result is likely explained by the high probability of low-ATE datasets in our prior, caused by the combination of additive noise and non-linearities with a squishing effect. This result highlights the need for future research in synthetic causal data-generation that creates strong treatment effects that persist across long causal chains.

**Dataset size** We also ablate upon Do-PFN's performance on datasets with between 5 and 2000 samples, observing in Figure 9 (center left) that Do-PFN's MSE in predicting CATE values decreases and converges in terms of variance as dataset size increases. This result is consistent with the generally strong performance of TabPFN on small-data Hollmann et al. (2023), which stems from TabPFN's pre-training objective: obtain maximally accurate predictions from synthetic datasets with an varied number of samples.

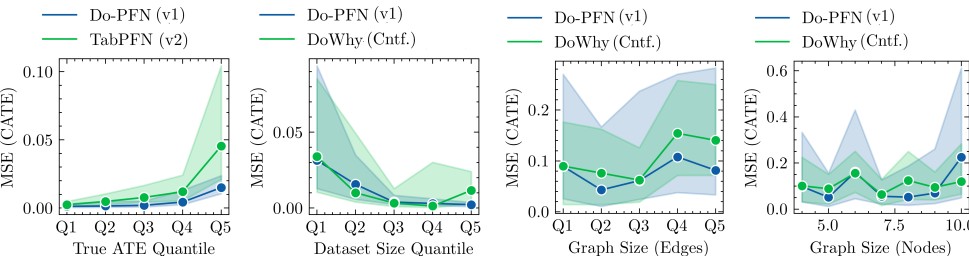

Figure 9: **In-distribution analysis**: Ablating the performance of Do-PFN across base rates of the average treatment effect (ATE), dataset sizes, and graph complexities. Do-PFN is robust to datasets with varied base rate ATE (left), improves in performance and consistency as dataset size grows (left center), and performs competitively with baselines that have access to the ground-truth graph structure as graph size and complexity grows (right).

**Graph size and complexity** We finally evaluate Do-PFN's performance across data generated from graphs of increasing complexity, sampling 500 datasets generated with graph structures consisting of 4 to 10 nodes and 2 to 43 directed edges. This result is visualized in Figure 9 (right), which shows that Do-PFN performs competitively to DoWhy (Cntf.), a baseline that has access to the ground-truth DAG across graphs of varying size and complexity. We note that while our synthetic data-generating mechanisms are relatively simple from a mathematical perspective, graph identification is a combinatorially hard problem, with the number of unique Directed Acyclic Graphs (DAGs) of 10 nodes reaching $4.17 \times 10^{18}$.

**Standard deviation of additive noise** We also highlight in Figure 10 (left) that the performance of Do-PFN decreases with an increase in the standard deviation of additive noise, which corresponds to a larger irreducible error. However, we also observe in Figure 10 (center-right) that Do-PFN's performance for different levels of additive noise seems to increase with dataset size. This means that the NMSE for datasets with a certain amount of additive noise can be reduced up to a certain extent with more data.

**Causal criteria** To investigate the robustness of Do-PFN and our CATE-estimation baselines with respect to the violation of the unconfoundedness assumption, we compare the performances on our

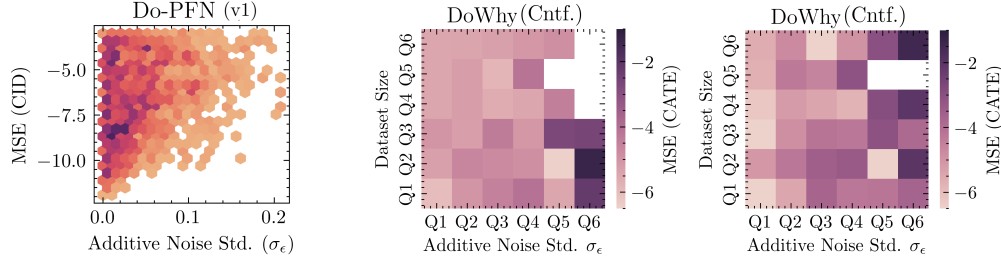

Figure 10: **In-distribution analysis (additive noise)**: Evaluation of Do-PFN's performance across different quantiles (Q1-Q5) of additive noise standard deviation. The density plot (left) shows that Do-PFN's performance decreases with (irreducible) additive noise. However, the heatmap (center) shows that for datasets with similar additive noise levels, Do-PFN's performance increases with dataset size. This effect is even stronger than for DoWhy (Cntf.).

synthetic case-studies that satisfy unconfoundedness (the "Observed Confounder" and "Backdoor-Criterion" case-studies) and that violate this assumption ("Observed Mediator", "Confounder + Mediator", "Unobserved Confounder", "Front-door Criterion"). The results in Figure 11 show that, while the performance of almost all methods degrades without unconfoundedness, Do-PFN maintains the strongest performance in both scenarios.

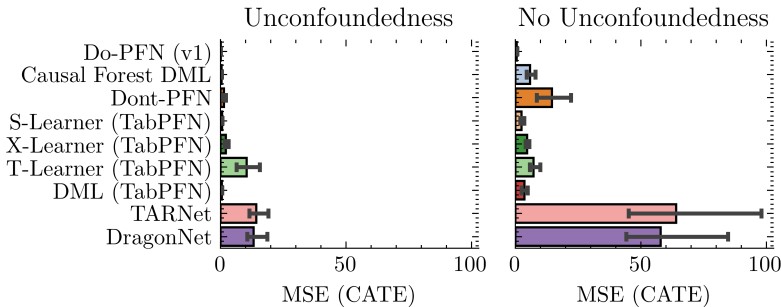

Figure 11: **Robustness to unconfoundedness**: Bar plots with 95% confidence intervals depicting the median MSE in CATE estimation when unconfoundedness (the treatment assignment is independent of the potential outcomes given the covariates) is fulfilled or violated. We find that while the performance of all methods degrades without unconfoundedness, Do-PFN maintains the strongest performance in both settings.

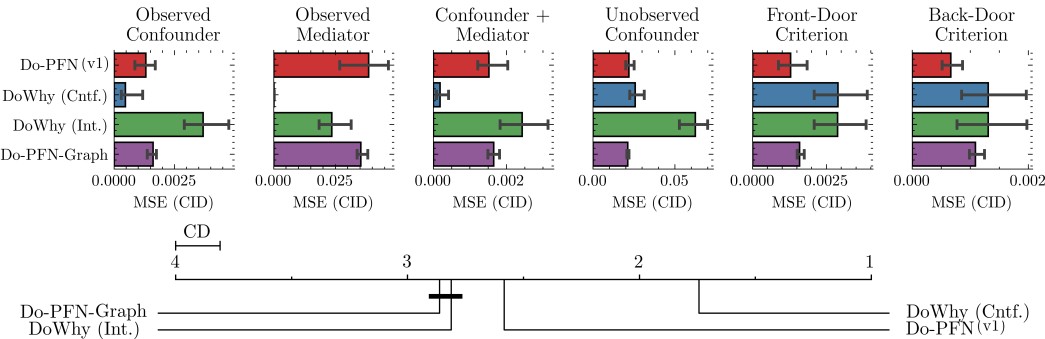

Figure 12: **Gold-standard comparison (CID)**: Box-plots and critical difference (CD) diagrams depicting distributions of normalized mean squared error (MSE) of Do-PFN and our "gold-standard" baselines in conditional interventional distribution (CID) estimation on our six synthetic case studies. Do-PFN significantly outperforms Do-PFN-Graph and DoWhy (Int.).

## E.3 Comparison to conceptual baselines

In this section, we compare Do-PFN to various conceptual baselines (i.e. baselines unlikely to be applied in the real-world) on our synthetic case studies.

**Gold standard baselines** First, we compare Do-PFN to a set of gold standard baselines which have access to the ground-truth DAG in order to further explore Do-PFN's ability to perform causal inference in settings where graph knowledge is required in order to correctly estimate causal effects, despite receiving no information regarding the causal structure. In Figure 12 we visualize bar-plots and critical difference (CD) diagrams of normalized-mean-squared-error in CID prediction. The results show that Do-PFN performs competitively with baselines Do-PFN-Graph which is pre-trained and evaluated on fixed graph structures, and DoWhy (Int.), in which a SCM is fit to the observational data and interventions are performed (without inferring noise terms, distinguising it from Do-Why (Cntf.)). In Figure 13, we observe a similar trend, except that in CATE estimation, Do-PFN rather performs competitively with Do-Why (Cntf.). We explain this result in a bias-decomposition (Figure 14) of the predictions of Do-PFN and Do-Why (Cntf.), which shows that although Do-PFN slightly overpredicts CIDs, its bias is consistent in the prediction of both potential outcomes and thus cancels out when predicting the CATE.

**Do-PFN variants** We also compare our proposed model with other variants, including Do-PFN-Short, Do-PFN-Graph, and Do-PFN-Mixed. In Figure 15, we observe that Do-PFN outperforms Do-PFN-Graph and Do-PFN-Short, which suggests that the number of pre-training datasets seen is a key factor in performance. We also observe that Do-PFN-Mixed, despite being pre-trained for only 24 hours, performs on par with our proposed model, suggesting that more complex noise distributions can aid in model convergance. This result also mirrors our out-of-distribution analysis, which shows that our model, which is pre-trained using only Gaussian additive noise, can generalize to other noise distributions.

## E.4 Common effect case study

We also provide an additional case-study, "Common-Effect" which perfectly represents a randomized controlled trial (RCT). We show in Figure 16 that only in this case, do traditional tabular predictors predict the CID with comparable performance. As discussed in Section 4.2, we hypothesize that this is largely due to the lack of distribution shift between the context and query set caused by the lack of a directed edge from the treatment to the covariate.

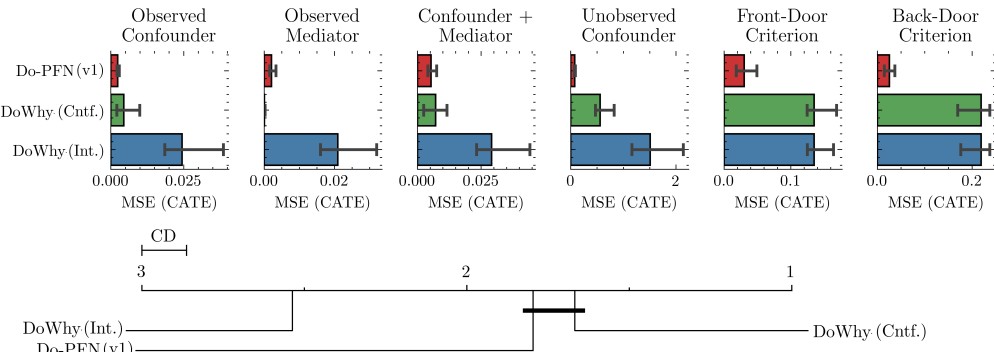

Figure 13: **Gold-standard comparison (CATE)**: Box-plots and critical difference (CD) diagrams depicting distributions of normalized mean squared error (MSE) of variants of Do-PFN and our baselines in conditional average treatment effect (CATE) estimation on our six synthetic case studies. Do-PFN-CATE outperforms DoWhy-CATE (Int.) and performs competitively with DoWhy-CATE (Cntf.).

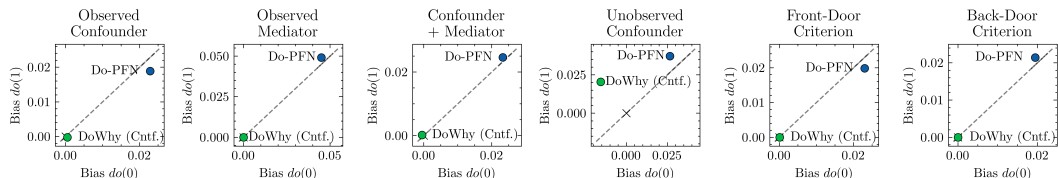

Figure 14: **Bias decomposition (CID)**: The median of the bias of DoWhy (Cntf.) and Do-PFN across 100 synthetic datasets for the interventions $do(0)$ and $do(1)$. The gold-standard DoWhy (Cntf.) maintains a bias very close to zero for all case-studies while Do-PFN has a small positive bias that takes almost the same value for do(0) and do(1).

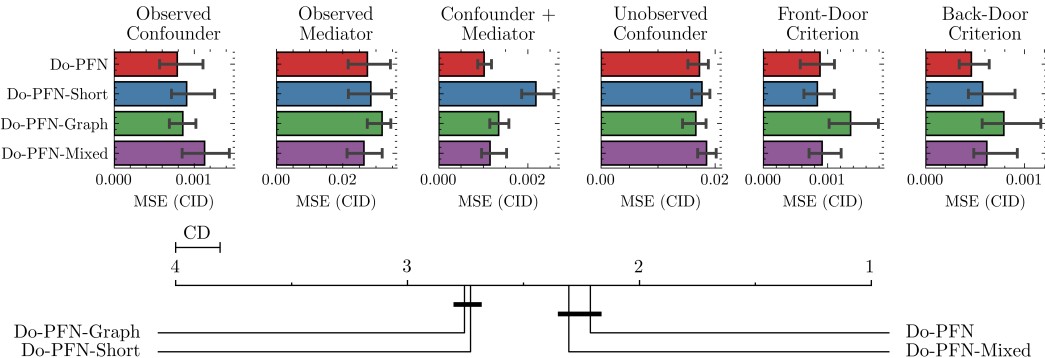

Figure 15: **Comparison of Do-PFN variants (CID)**: Box-plots and critical difference (CD) diagrams depicting distributions of normalized mean squared error (MSE) of Do-PFN variants in conditional interventional distribution (CID) estimation on our six synthetic case studies. Do-PFN significantly outperforms other variants except Do-PFN-Mixed, which achieves statistically similar performance in half the pre-training time.

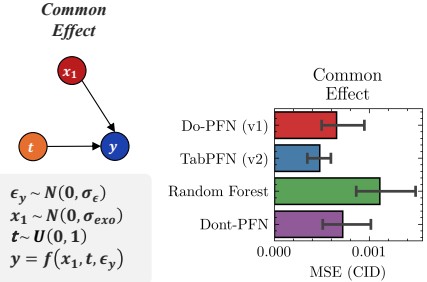

Figure 16: **Common effect case study**: Visualization of graph structure and structural equations (left) for our "Common-Effect" case study, as well as box plots depicting distributions of normalized mean squared error (MSE) of Do-PFN compared to regression baselines in conditional interventional distribution prediction. Regression baselines perform similarly to Do-PFN, as the intervention does not cause a distribution shift between $\mathcal{D}^{ob}$ and $\mathcal{D}^{in}$.

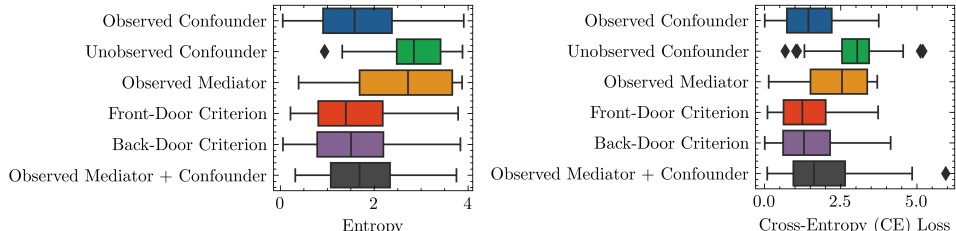

Figure 17: **Uncertainty quantification**: Cross-entropy (CE) loss (right) and entropy (left) of Do-PFN's bar distribution output. Do-PFN is highly uncertain on the "Unobserved Confounder" case study due to unidentifiability. Do-PFN also shows high uncertainty on the "Observed Mediator" case study, which we argue is due to its only exogenous term being a binary variable, causing the continuous effect in the outcome to only come from additive noise.)

## E.5 Uncertainty

Next, we explore Do-PFN's uncertainty calibration, by visualizing the prediction interval coverage probability (PICP) in Figure 9. A PICP curve equal to the 45-degree diagonal corresponds to a model consistently yielding prediction intervals with exactly the desired coverage. Being above the diagonal corresponds to under-confident and being below the diagonal to over-confident prediction intervals. First, we observe that Do-PFN's uncertainty is slightly under-confident for theoretically identifiable case studies (the full set of calibration plots can be found in Appendix E.5). In the "Unobserved Confounder" case study, the model's high uncertainty is reflected by a relatively large entropy in its output distribution However, our PICP results show that the model's uncertainty for this case study is correctly calibrated.

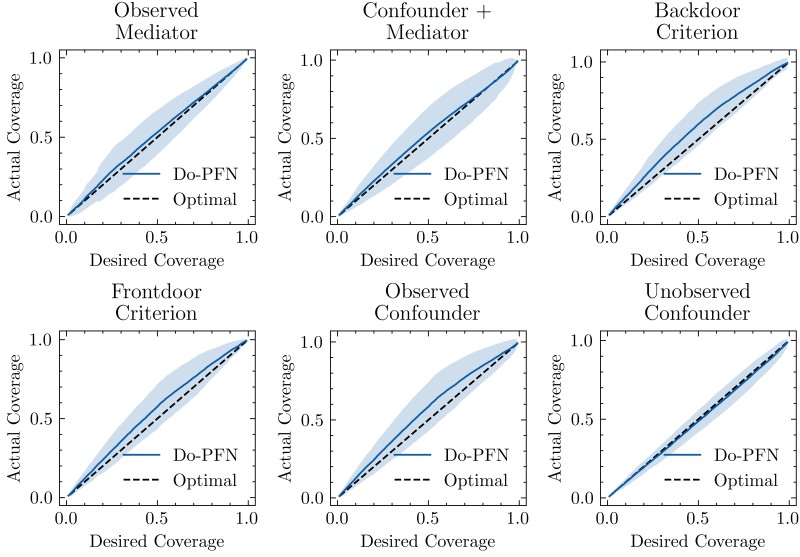

Figure 18: **Uncertainty calibration**: Prediction interval coverage probability (PICP) plots for the "Observed Mediator", "Confounder + Mediator", "Backdoor Criterion" and "Frontdoor Criterion" cases. The solid blue line shows the coverage and standard deviation achieved by Do-PFN, spanning desired probabilities from 0 to 1. The dashed line represents the ideal calibration achievable with access to the ground-truth CID. Do-PFN is slightly under-confident for identifiable case studies, and, crucially, correctly unconfident for the "unobserved confounder" case.

## E.6 Critical Difference Plots

To compare model performances across multiple datasets in a statistically rigorous way, we employ the `autorank` package (Herbold, 2020), which performs a non-parametric Friedman test followed by pairwise post-hoc analysis using the Nemenyi test. The resulting *critical difference (CD) plots* visualize average ranks of models across datasets: models that are not significantly different at $\alpha = 0.05$ are connected by horizontal bars. Lower ranks correspond to better performance. This approach allows us to jointly assess both the magnitude and statistical significance of performance differences across all case studies.

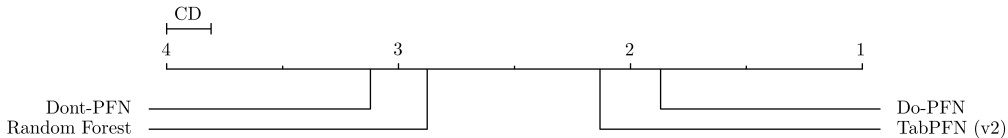

Figure 19: **Critical difference (CID)**: Critical difference (CD) diagram depicting distributions of normalized mean squared error (MSE) of Do-PFN and our regression baselines in conditional interventional distribution (CID) prediction. Across our six causal case studies, Do-PFN achieves statistically significant improvementsover regression baselines, showing that our pre-training objective is effective for predicting CIDs.

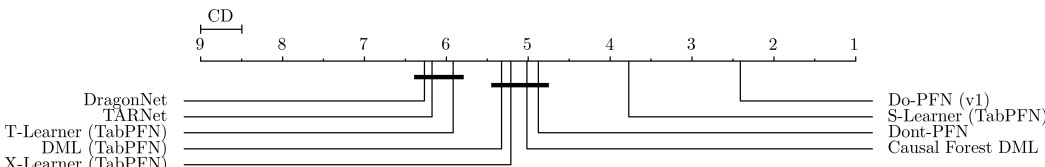

Figure 20: **Critical difference (ATE)**: Critical difference (CD) diagram depicting distributions of normalized mean squared error (MSE) of Do-PFN and our regression baselines in average treatment effect (ATE) estimation. Across our six causal case studies, Do-PFN significantly outperforms our meta-learner, double machine learning, and deep-learning based baselines.

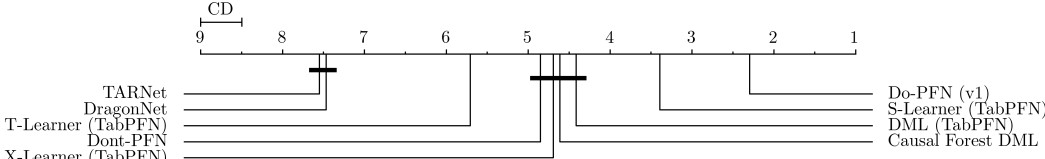

Figure 21: **Critical difference (CATE)**: Critical difference (CD) diagram depicting distributions of normalized mean squared error (NMSE) of Do-PFN-CATE and our causal baselines in conditional average treatment effect (CATE) estimation. Across our six causal case studies, Do-PFN significantly outperforms our meta-learner, double machine learning, and deep-learning based baselines.

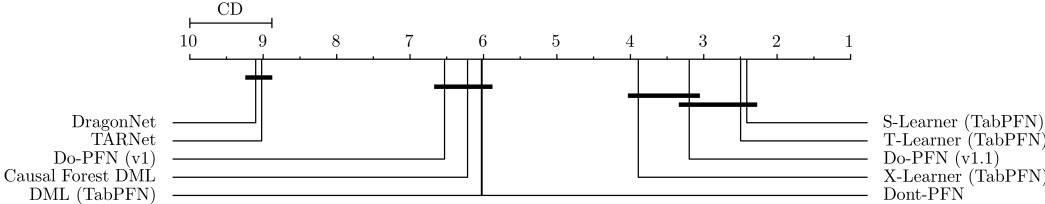

Figure 22: **Critical difference (RealCause)**: Critical difference (CD) diagram depicting distributions of normalized mean squared error (NMSE) of Do-PFN-CATE and our causal baselines in conditional average treatment effect (CATE) estimation on the RealCause benchmark. Across all four dataset groups, Do-PFN maintains competitive performance with our meta-learner, double machine learning, and deep-learning based baselines.

