# OpenReview forum: "Do-PFN: In-Context Learning for Causal Effect Estimation"
_NeurIPS.cc/2025/Conference — NeurIPS 2025 spotlight_

### Official Review · Reviewer_Ww3g · 2025-06-10

**Clarity:** 3
**Significance:** 2
**Originality:** 2
**Rating:** 5
**Confidence:** 4

**Summary:**

This paper proposes a novel meta-learning approach for causal effect estimation that pre-trains a Prior-data fitted network (PFN) on synthetic observational and interventional data drawn from various common structural causal models. Differing from the inspired work of TabPFN, designed for standard prediction tasks on tabular datasets given in-context training sets, Do-PFN performs amortized causal inference by outputting a conditional intervention distribution given an observational dataset, interventional covariates and an intervened treatment variable. The trained model is evaluated on synthetic data from six common graph structures requiring Do-PFN to perform front door and back door adjustment. It is shown that the learned method performs comparably to baseline algorithms that access the true causal graph in terms of CID and CATE and that it outperforms other baselines without this assumption. Ablation demonstrates the method performs well even on small datasets and is robust to magnitude of ATE and graph sizes.

**Questions:**

Some questions were raised above.

1. The ACTIVA method presented in Sauter et al. seems to have many similarities with Do-PFN. While I understand that the output of ACTIVA produces the shift in distribution for all variables after performing and intervention, wouldn’t this also inherit the shift in outcome variables distribution which Do-PFN is predicting? I am curious what are the primary differences that makes Do-PFN a stronger approach than ACTIVA, and accordingly where your paper's main source of novelty lies.

**Ethical Concerns:**

["NO or VERY MINOR ethics concerns only"]

**Final Justification:**

I am much more confident following the rebuttal from the authors that Do-PFN can be used as a practical tool for the accurate estimation of causal effects without prior knowledge of the causal graph. As far as I am aware, using in-context learning approaches for causal effect estimation is quite novel and quite attractive, and while the work may have some limitations in terms of the intervention types they accept and still questionable prior matching, I believe it is a great starting place for considering a new look on causal inference.

**Limitations:**

Yes

**Quality:**

3

**Strengths And Weaknesses:**

Strengths:
1. Applying the popular PFN to the task of causal inference is an attractive extension.
2. Presents a reasonable framework for performing causal inference without knowledge of the causal graph or prior interventions, increasing practical utility.
3. Quite interesting that Do-PFN can capture uncertainty arising from unidentifiability of causal effects.
4. Paper is very well-written

Weaknesses:
1. Do-PFN is entirely pretrained on synthetic data from a very constrained set of possible SCMs as explained in Appendix B. Given evaluation is performed on the exact same class of synthetic SCMs, it is not surprising that the performance would be quite strong. However, since there is no discussion of why this synthetic prior over SCMs would capture real-world causal systems, it is unclear whether this model would at all useful in practice. Additional experiments demonstrating that this method of prior fitting can lead to good real-world performance seems essential.
2. Paper lacks a clear understanding of where and to what extent uncertainty arises in the model’s predictions. Ie. why is it the case that uncertainty is under confident for theoretically identifiable cases?
3. This work considers only the class of single-node binary interventions, which is restrictive in practice. A consideration of multi-node continuous interventions would be much stronger.

---

> ### Author Rebuttal · Authors · 2025-07-29
>
> # General Response
> ----
>
> All reviewers acknowledge that our paper presents a promising approach, but also raise the question of prior mismatch, especially when analyzing real-world data. While we believe it is crucial for a new method to study synthetic data with known ground truth, we agree that analyzing real world data is important. Therefore, we have meanwhile performed an analysis of Do-PFN's performance on **six datasets based on real-world data**, with promising results that we detail next:
>
> **Known Graph:** First, we evaluate on the Amazon Sales and Law School Admissions datasets where the causal graph is known (please refer to the .zip file in the supplementary material attached to the original submission for details):
>
> ## Median MSE (CATE) - Known Graph
> |Method|Law School|Amazon Sales|
> |:------------------------|:------------------|:------------------|
> |Causal Forest DML|0.157 ± 0.056|0.136 ± 0.279|
> |Causal Forest DML (HPO)|0.519 ± 0.080|0.579 ± 0.316|
> |DML (TabPFN)|0.283 ± 0.067|**0.037 ± 0.169**|
> |Do-PFN (v1)|**0.029 ± 0.010**|**0.038 ± 0.023**|
> |Do-PFN (v1.1)|**0.067 ± 0.043**|0.209 ± 0.135|
> |Dont-PFN|**0.103 ± 0.046**|0.313 ± 0.158|
> |DragonNet|0.371 ± 0.035|0.579 ± 0.316|
> |S-Learner (HPO)|0.123 ± 0.045|0.233 ± 0.246|
> |S-Learner (TabPFN)|0.161 ± 0.051|0.568 ± 0.303|
> |TARNet|0.372 ± 0.119|0.579 ± 0.316|
> |T-Learner (HPO)|0.151 ± 0.095|**0.017 ± 0.072**|
> |T-Learner (TabPFN)|0.161 ± 0.048|0.579 ± 0.316|
> |X-Learner (HPO)|0.519 ± 0.080|0.579 ± 0.316|
> |X-Learner (TabPFN)|0.134 ± 0.047|0.579 ± 0.316|
>
> We find that Do-PFN (v1) achieves the strongest performance on Law School and is not significantly worse than the best method on Amazon Sales, a dataset with a complex mediation structure.
>
> **RealCause:** We also evaluate on four new datasets from the RealCause benchmark [5], using a version of Do-PFN (which we call v1.1) pre-trained on larger graphs with up to 60 nodes.
>
> We observe that Do-PFN-v1.1’s extended pre-training leads to strong performance among competitive baselines on three out of four RealCause datasets. We note that these datasets perfectly satisfy unconfoundedness, giving our baselines a fundamental advantage against Do-PFN. Due to space constraints we refer the Table “**Median MSE (CATE) - RealCause**” in our general response to reviewer **Fczh** for the precise MSE values.
>
> These results show Do-PFN to nevertheless be competitive with a strong set of CATE baselines, suggesting the strong transfer of Do-PFN's synthetic pre-training to complex, real-world data.
>
>
> # Response to Reviewer Ww3g
> ----
>
> Thank you for your feedback and interesting questions!
>
> > Paper lacks a clear understanding of where and to what extent uncertainty arises in the model’s predictions.
>
> We thank you for encouraging us to further develop our theory regarding uncertainty!
>
> The posterior predictive distribution that Do-PFN approximates can be factorized as follows: $p(y|x, do(t), D) = \int p(y|x, do(t), \psi) p(\psi|D) d\psi$, where $y$ denotes the outcome, $x$ a covariate vector, $\psi$ an SCM,  and $D$ an observational dataset.
>
> Let’s also consider the observational equivalence relation $\psi_1 \sim \psi_2 \iff p(y,x,t|\psi_1) = p(y,x,t|\psi_2)$, where a selector $r(\psi)$ maps any SCM $\psi$ onto the representative of its equivalence class $[\psi]$. Subsequently, this yields:
>
> $p(y|x, do(t), D) = \int \int p(y|x, do(t), \psi) p(\psi|r) p(r|D) dr d\psi$
>
> This factorization allows to distinguish three types of uncertainty:
>
> - First, $p(y|x, do(t), \psi)$ represents the uncertainty in the conditional interventional distribution of the outcome $y$ assuming a **fixed** SCM $\psi$; all uncertainty about $y$ in $p(y|x, do(t), \psi)$ is caused by the noise distributions in the SCM $\psi$ and thus represents a form of aleatoric uncertainty.
>
> - Second, $p(\psi|r)$ is the distribution over observationally equivalent SCMs assuming we are in equivalence class $[r]$, and thus represents uncertainty due to potential unidentifiability of $\psi$ from observational data.
>
> - Finally, the distribution $p(r|D)$ quantifies the epistemic uncertainty about the Markov equivalence class of a ground truth SCM caused by a finite observational dataset $D$.
>
> We show, in a newly added section of our paper, that for infinite observational data the uncertainty in $p(r|D)$ vanishes completely, but the other sources of uncertainty remain. More specifically, when the observational data $D$ comes from a ground-truth SCM $\psi_0$, the posterior conditional interventional distribution $p(y|x, do(t), D)$ converges to the posterior $p(y|x,do(t),[\psi_0])$ where $[\psi_0]$ denotes the set of all observationally equivalent SCMs.
>
> More formally:
>
> $$p(y|x, do(t),D) \rightarrow p(y|x,do(t),[\psi_0]) =  \int p(y|x, do(t), \psi) p(\psi|[\psi_0]) d\psi, \quad \text{for}\  |D| \rightarrow \infty.$$
>
> We formulate all those arguments in a rigorous way using measure theory and Doob’s theorem in a new theoretical section in the appendix of our manuscript.
>
> > Why is it the case that uncertainty is under-confident for theoretically identifiable cases?
>
> This is a very interesting question!
>
> As we show in Proposition 1, we minimize the **forward** KL-divergence $D_{KL}[p(y|x, do(t), D)||q(y|x,do(t), D)]$ between the posterior $p(y|x, do(t), D)$ and the model $q(y|x,do(t), D)$. This leads to $q$ being mass covering, i.e. overdispersed—especially when $q$ is a suboptimal approximation of $p$ [2,3]. We parametrize $q$ as a bar distribution, with fixed positions of the bars following TabPFN [4]. In the identifiable case the posterior is more concentrated leading to a slightly larger discrepancy between the posterior $p$ and the model $q$ potentially due to discretization error. This difference manifests itself in Do-PFN being **slightly** underconfident (please see our calibration curves), rather than overconfident due to the nature of the forward KL-divergence.
>
> Please also note that underconfidence in case of approximation error is typically considered a very desirable property [3].
>
> > While I understand that the output of ACTIVA produces the shift in distribution for all variables after performing and intervention, wouldn’t this also inherit the shift in outcome variables distribution which Do-PFN is predicting? I am curious what are the primary differences that makes Do-PFN a stronger approach than ACTIVA, and accordingly where your paper's main source of novelty lies.
>
> Thank you for the opportunity to point out the differences between Do-PFN and the contemporaneous ACTIVA method. We summarize some of the major differences below:
>
> - **Task:** DoPFN is trained to learn how to predict the effect of a binary intervention on a univariate outcome variable, which is a fundamental yet practically relevant and widely researched task in causal inference with a broad set of available baselines and benchmarks [5]. ACTIVA considers the problem of learning the effect of multi-variable interventions on the multivariate distribution of all variables in an SCM simultaneously.
>
> - **Training data:** ACTIVA is trained on data from fixed-sized, purely linear additive causal models (without any nonlinearities) and the SERGIO simulator (with 11 nodes) with knockout interventions. Causal sufficiency is assumed, i.e. covariates in an SCM are observed. In contrast, we vary the nonlinearities, graph sizes and noise distributions and also include unobservable variables. Conceptually, it would be possible to marginalize the output distribution of an ACTIVA-model after conditioning on only a single intervention in order to obtain predictions for a single variable. However, this would not bridge the gap between vastly different pre-training setups.
>
> - **Model:** Do-PFN learns a univariate posterior predictive and employs the same architecture as TabPFN [4], as well as the same histogram-based posterior approximation, which has been found to be highly effective for predictive tasks. ACTIVA learns a multivariate posterior and employs a $\beta$-VAE with Vamp prior plus a mixture-of-Gaussian parametrization of the conditional interventional distributions and is based on transformer encoders and decoders with alternating row- and column attention.
>
> - **Evaluation:** We compare Do-PFN against established methods for causal effect estimation besides observational baselines. ACTIVA is evaluated in terms of similarity of the predicted distribution compared to the ground truth interventional distribution (for instance evaluated in terms of MMD); we investigate the **predictive performance** of Do-PFN on specific causal graphs and semi-synthetic data.
>
> > This work considers only the class of single-node binary interventions, which is restrictive in practice. A consideration of multi-node continuous interventions would be much stronger.
>
> We focus on this specific task since it is one of the most researched problems in causal inference, see e.g. [5]. We believe that, due to its relevance, this warrants studies that fully evaluate the performance of different methods on synthetic and semi-real benchmarks, as well as comparisons against numerous existing baselines. It’s a good starting point.
>
> ## References
>
> [1] Neal et al. "Realcause: Realistic Causal Inference Benchmarking." arXiv 2020.
>
> [2] Murphy, Kevin P. "Probabilistic machine learning: Advanced topics." MIT press, 2023.
>
> [3] McNamara, et al. "Sequential monte carlo for inclusive kl minimization in amortized variational inference." AISTATS 2024.
>
> [4] Hollmann, Noah, et al. "TabPFN: A Transformer That Solves Small Tabular Classification Problems in a Second." NeurIPS 2022.
>
> [5] Yao, Liuyi, et al. "A survey on causal inference." ACM Transactions on Knowledge Discovery from Data (TKDD) 15.5 (2021): 1-46.

---

> > ### Comment · Reviewer_Ww3g · 2025-08-01
> > **Score Adjustment**
> >
> > Thank you for your very well written response and additional experimental results. In light of these updates, I am happy to increase my evaluation score. I look forward to seeing if this methodology can be extended to consider different types of interventions in the future as well as further convincing results on real-world data.

---

### Official Review · Reviewer_bczT · 2025-06-27

**Clarity:** 3
**Significance:** 3
**Originality:** 3
**Rating:** 5
**Confidence:** 3

**Summary:**

This paper proposes Do-PFN, a transformer-based architecture trained via meta-learning to perform causal effect estimation from observational data. Building upon the prior-data fitted network (PFN) framework, the authors pre-train Do-PFN on a large number of synthetic datasets simulated from structural causal models (SCMs), including both observational and interventional data. The model is trained to predict interventional outcomes given only observational data and a target intervention. The authors evaluate Do-PFN across a suite of synthetic benchmarks reflecting canonical causal inference challenges (e.g., back-door, front-door, and unobserved confounding), demonstrating that it accurately predicts conditional interventional distributions (CIDs) and conditional average treatment effects (CATEs). Notably, Do-PFN achieves competitive or superior performance compared to baselines that rely on access to the ground-truth causal graph, despite requiring only observational data.

**Questions:**

Have you considered evaluating Do-PFN on real-world benchmarks for causal effect estimation (e.g., IHDP, Twins)? Even if ground truth is not perfect, such comparisons would help assess real-world transfer and robustness.

How sensitive is Do-PFN to mismatches between the synthetic prior used during training and the test-time SCMs?

The paper claims Do-PFN performs implicit adjustment akin to back-door or front-door criteria. Could the authors include examples or diagnostics to confirm that the model performs appropriate adjustment?

Do-PFN is pre-trained for up to 48 hours on a single GPU. How does the training time and inference speed compare to existing causal inference methods, especially in settings with larger datasets or real-time requirements?

**Ethical Concerns:**

["NO or VERY MINOR ethics concerns only"]

**Final Justification:**

This is overall a good submission. The author(s) engaged during rebuttal, so I am happy to recommend this paper for acceptance.

**Limitations:**

Yes

**Quality:**

3

**Strengths And Weaknesses:**

The paper's main strengths lie in its originality, empirical rigor, and the clarity of its contributions. It presents a compelling case for applying PFNs to causal inference, offering a novel approach to a long-standing challenge: estimating causal effects in the absence of a known causal graph. The proposed method is grounded in a well-defined meta-learning framework and makes use of amortized inference via synthetic SCMs. The authors offer both theoretical motivation and empirical validation. The paper is overall well-written

However, the paper has a few limitations. While the use of synthetic data is well-justified in this context, it is unclear how well Do-PFN generalizes to real-world settings. The paper briefly acknowledges this issue, but does not provide empirical results or strong evidence of robustness under real-world distribution shifts or misspecified priors. The lack of real data experiments and the reliance on synthetic priors with assumed coverage over plausible real-world SCMs is the most substantial weakness. While the paper claims Do-PFN learns to “implicitly” adjust for confounding (e.g., back-door/front-door), it would benefit from deeper analysis.

---

> ### Author Rebuttal · Authors · 2025-07-29
>
> # General Response
> ----
>
> All reviewers acknowledge that our paper presents a promising approach, but also raise the question of prior mismatch, especially when analyzing real-world data. While we believe it is crucial for a new method to study synthetic data with known ground truth, we agree that analyzing real world data is important. Therefore, we have meanwhile performed an analysis of Do-PFN's performance on **six datasets based on real-world data**, with promising results that we detail next:
>
> **Known Graph:** First, we evaluate on the Amazon Sales and Law School Admissions datasets where the causal graph is known (please refer to the .zip file in the supplementary material attached to the original submission for details):
>
> ## Median MSE (CATE) - Known Graph
> |Method|Law School|Amazon Sales|
> |:------------------------|:------------------|:------------------|
> |Causal Forest DML|0.157 ± 0.056|0.136 ± 0.279|
> |Causal Forest DML (HPO)|0.519 ± 0.080|0.579 ± 0.316|
> |DML (TabPFN)|0.283 ± 0.067|**0.037 ± 0.169**|
> |Do-PFN (v1)|**0.029 ± 0.010**|**0.038 ± 0.023**|
> |Do-PFN (v1.1)|**0.067 ± 0.043**|0.209 ± 0.135|
> |Dont-PFN|**0.103 ± 0.046**|0.313 ± 0.158|
> |DragonNet|0.371 ± 0.035|0.579 ± 0.316|
> |S-Learner (HPO)|0.123 ± 0.045|0.233 ± 0.246|
> |S-Learner (TabPFN)|0.161 ± 0.051|0.568 ± 0.303|
> |TARNet|0.372 ± 0.119|0.579 ± 0.316|
> |T-Learner (HPO)|0.151 ± 0.095|**0.017 ± 0.072**|
> |T-Learner (TabPFN)|0.161 ± 0.048|0.579 ± 0.316|
> |X-Learner (HPO)|0.519 ± 0.080|0.579 ± 0.316|
> |X-Learner (TabPFN)|0.134 ± 0.047|0.579 ± 0.316|
>
> We find that Do-PFN (v1) achieves the strongest performance on Law School and is not significantly worse than the best method on Amazon Sales, a dataset with a complex mediation structure.
>
> **RealCause:** We also evaluate on four new datasets from the RealCause benchmark [5], using a version of Do-PFN (which we call v1.1) pre-trained on larger graphs with up to 60 nodes:
>
> ##  Median MSE (CATE) - RealCause
> |Method|LalondeCPS|LalondePSID|IHDP|ACIC2016|
> |:-------------------|:------------------|:------------------|:------------------|:------------------|
> |Causal Forest DML|0.114 ± 0.014|0.016 ± 0.002|0.024 ± 0.001|0.005 ± 0.001|
> |DML (TabPFN)|0.067 ± 0.009|0.013 ± 0.001|0.025 ± 0.002|0.008 ± 0.004|
> |Do-PFN (v1)|0.067 ± 0.006|0.022 ± 0.003|0.031 ± 0.002|0.033 ± 0.004|
> |Do-PFN (v1.1)|**0.039 ± 0.004**|**0.008 ± 0.001**|0.021 ± 0.001|0.019 ± 0.011|
> |Dont-PFN|0.049 ± 0.004|0.011 ± 0.001|0.032 ± 0.001|0.036 ± 0.003|
> |DragonNet|0.064 ± 0.006|0.020 ± 0.003|0.543 ± 0.154|0.097 ± 0.035|
> |S-Learner (HPO)|0.068 ± 0.010|0.015 ± 0.001|0.022 ± 0.002|0.003 ± 0.001|
> |S-Learner (TabPFN)|0.053 ± 0.007|0.013 ± 0.003|**0.019 ± 0.001**|**0.002 ± 0.001**|
> |T-Learner (HPO)|**0.044 ± 0.003**|**0.007 ± 0.001**|0.023 ± 0.002|0.002 ± 0.001|
> |T-Learner (TabPFN)|**0.042 ± 0.002**|**0.007 ± 0.001**|**0.020 ± 0.001**|**0.002 ± 0.001**|
> |TARNet|0.064 ± 0.006|0.020 ± 0.003|0.517 ± 0.147|0.092 ± 0.032|
> |X-Learner (HPO)|0.069 ± 0.008|0.020 ± 0.003|0.155 ± 0.033|0.048 ± 0.016|
> |X-Learner (TabPFN)|0.083 ± 0.010|0.015 ± 0.005|**0.019 ± 0.001**|**0.001 ± 0.000**|
>
> We observe that Do-PFN-v1.1’s extended pre-training leads to strong performance among competitive baselines on three out of four RealCause datasets. We note that these datasets perfectly satisfy unconfoundedness, giving our baselines a fundamental advantage against Do-PFN. These results show Do-PFN to nevertheless be competitive with a strong set of CATE baselines, suggesting the strong transfer of Do-PFN's synthetic pre-training to complex, real-world data.
>
>
> # Response to Reviewer bczT
> ----
>
> We thank you for your positive feedback and very meaningful questions!
>
> > How sensitive is Do-PFN to mismatches between the synthetic prior used during training and the test-time SCMs?
>
> To address this,  in addition to the strong real-world performance we observe above, we also perform several ablation studies regarding distribution mismatch. We also refer to our response to reviewer **Fczh**, where we provide OOD results in terms of graph size, and a theoretical argument regarding prior mismatch.
>
> Our new out-of-distribution analysis shows the performance of Do-PFN on in-distribution samples from its prior, compared to out-of-distribution samples in terms of 1) graph size, 2) non-linearity type, and 3) noise distributions.
>
> ## Median MSE - Out of Distribution
>
> |OOD Type|Do-PFN (v1)|% Change|
> |:--------------------|:--------------------|:-----------|
> |In Distribution|**0.0053 ± 0.0008**|0.0%|
> |ELU Non-Linearity|0.0033 ± 0.0004|-38.2%|
> |Post Non-Linearity|0.0057 ± 0.0006|7.0%|
> |Sin Non-Linearity|0.0066 ± 0.0007|24.6%|
> |Student-t Noise|0.0050 ± 0.0006|-4.6%|
> |Laplacian Noise|0.0060 ± 0.0008|13.3%|
> |Gumbel Noise|0.0059 ± 0.0007|10.9%|
> |High Gaussian Noise|0.0212 ± 0.0006|301.4%|
>
> We find that Do-PFN is most robust to changes in the noise distribution and with respect to post-non-linearities, with its MSE in CID prediction changing by a magnitude of less than 0.001. Interestingly in some cases, such as the “ELU Non-Linearity” ablation, going out of distribution results in an increase in performance, possibly since ELU shares functional similarities to the ReLU activation which is represented in our prior.
>
> > How does the training time and inference speed compare to existing causal inference methods in settings with larger datasets or real-time requirements?
>
> We thank you for the opportunity to discuss the relationship between training- and inference time of Do-PFN. Do-PFN is a foundation model of 7.3 million parameters (small compared to current LLMs, but large, compared to traditional causality methods). It is pre-trained for up to 48 hours, but its pre-training time is incurred **once** up-front. When applying DoPFN to a new dataset, predictions can be obtained in a single forward pass.
>
> We now provide inference time results, calculating the median time in seconds of the *.fit* and *.predict* functions of Do-PFN and our baselines. We also provide versions of several baselines with HPO for 60 seconds per underlying base model, and run deep-learning-based methods on an RTX-2080 GPU.
> ## Median Train/Optimize + Inference Time (s)  - RealCause
>
> |Method|LalondeCPS|LalondePSID|IHDP|ACIC2016|
> |:-----------------------|:------------------|:------------------|:------------------|:------------------|
> |Causal Forest DML|8.075 ± 0.090|8.014 ± 0.083|6.839 ± 0.051|19.070 ± 0.239|
> |DML (TabPFN) GPU|12.376 ± 0.035|12.441 ± 0.094|11.740 ± 0.029|22.807 ± 0.271|
> |Do-PFN (v1) CPU|2.699 ± 0.047|2.324 ± 0.131|**0.699 ± 0.009**|**2.734 ± 0.179**|
> |Do-PFN (v1) GPU|**1.029 ± 0.098**|**1.311 ± 0.162**|1.270 ± 0.143|**1.060 ± 0.160**|
> |Do-PFN (v1.1) CPU|3.010 ± 0.201|2.188 ± 0.013|**0.655 ± 0.004**|4.474 ± 2.042|
> |Do-PFN (v1.1) GPU|**1.011 ± 0.002**|**0.990 ± 0.004**|0.962 ± 0.001|**1.035 ± 0.036**|
> |Dont-PFN GPU|2.916 ± 0.169|2.437 ± 0.097|**0.769 ± 0.005**|3.258 ± 0.125|
> |DragonNet GPU|67.469 ± 14.605|124.642 ± 10.753|6.791 ± 0.013|6.378 ± 0.249|
> |S-Learner (HPO) |60.160 ± 0.072|60.373 ± 0.117|60.066 ± 0.012|60.498 ± 0.206|
> |S-Learner (TabPFN) GPU|**2.001 ± 0.004**|**1.927 ± 0.012**|1.791 ± 0.002|4.694 ± 0.105|
> |T-Learner (TabPFN) GPU|3.473 ± 0.009|3.573 ± 0.168|3.309 ± 0.005|7.378 ± 0.202|
> |TARNet GPU|58.258 ± 21.274|119.821 ± 14.448|6.804 ± 0.008|6.728 ± 0.293|
> |X-Learner (HPO) |120.240 ± 0.094|120.172 ± 0.102|120.060 ± 0.007|120.201 ± 0.050|
> |X-Learner (TabPFN) GPU|13.555 ± 0.239|13.469 ± 0.216|12.886 ± 0.037|31.158 ± 2.123|
> |Dataset Shape|2500,5|1340,5|375,7|2405,50|
>
> In our runtime analysis, Do-PFN (run on CPU and GPU) has a faster inference speed than all other baselines. Do-PFN outperforms other baselines because there is no conventional model-fitting; and is slightly faster than the TabPFN meta-learning approaches (when both methods use a GPU), since Do-PFN has less parameters and does not do ensembling.
>
> > While the paper claims Do-PFN learns to “implicitly” adjust for confounding (e.g., back-door/front-door), it would benefit from deeper analysis.
>
> While we empirically find that Do-PFN works very well across diverse causal challenges, on both carefully controlled synthetic setups (including front-door and back-door cases) as well as real world data, the precise inner workings of PFNs in general, and thus also Do-PFN, are rather unclear. We consider it unlikely that Do-PFN works analogously to classical multi-step approaches that would, e.g., perform causal discovery, then determine estimands, and finally carry out the appropriate adjustment, but the precise in-context-learning dynamics of Do-PFN remain to be explored.
>
> In the field of LLMs, mechanistic interpretability tools have become popular and their applications to PFNs, and Do-PFN in particular, might allow to systematically trace specific procedures or algorithms Do-PFN utilizes internally.
> Based on your question, we will expand on this as a key area of future work in the discussion section.
>
> ## References
>
> [1] Neal et al. "Realcause: Realistic Causal Inference Benchmarking." arXiv 2020.
>
> [2] Bereska, Leonard, and Stratis Gavves. "Mechanistic Interpretability for AI Safety-A Review." TMLR 2024.

---

> > ### Comment · Reviewer_bczT · 2025-08-01
> >
> > Thank you for your reply. My score remains unchanged.

---

### Official Review · Reviewer_HtXo · 2025-06-29

**Clarity:** 4
**Significance:** 3
**Originality:** 3
**Rating:** 5
**Confidence:** 4

**Summary:**

This paper extends Prior-data Fitted Networks (PFNs) to causal effect estimation on tabular data, introducing Do-PFNs. The key innovation is training PFNs on synthetic datasets spanning diverse causal structures. Do-PFNs achieve performance comparable to methods with access to ground-truth causal graphs while substantially outperforming baselines that lack this information.

**Questions:**

I might increase the scores based on authors response to my questions below:
- Intuitions on why the model is underconfident in identifiable cases?
- Can you run experiment estimating ATE and compare with the same baselines?
- Possibility of including more variables (e.g. number of confounders in your experiment)?

**Ethical Concerns:**

["NO or VERY MINOR ethics concerns only"]

**Final Justification:**

The authors have addressed my questions properly and the additional experiments showed evidence of strength of their method.

**Limitations:**

Yes

**Quality:**

3

**Strengths And Weaknesses:**

*Strengths*:
- The theoretical analysis provides valuable insights into Do-PFN's empirical performance, offering principled understanding of why the approach works.
- The experiments convincingly demonstrate that Do-PFN learns to estimate causal effects rather than merely fitting observational outcome distributions, as evidenced by comparisons with Dont-PFN.
- The experimental evaluation is comprehensive, featuring: (1) diverse causal scenarios covering various structural assumptions, (2) comparison against strong baseline methods, and (3) thorough ablation studies that isolate key design choices.

*Weaknesses*:
- The experimental evaluation is limited to simulated datasets with at most 5 variables, which may not capture the complexity and dimensionality of real-world causal inference problems.
- The exclusion of Average Treatment Effect (ATE) estimation from the main experimental results is unexplained, despite ATE being a fundamental quantity of interest in causal inference.
- The computational efficiency appears questionable, deploying a 7.3 million parameter model for datasets with only 5 variables raises concerns about scalability and whether such model complexity is warranted for these problem sizes.
- The paper lacks theoretical or empirical analysis explaining two key behavioral patterns: why the model exhibits underconfidence in identifiable cases and why it performs well specifically on small datasets.

---

> ### Author Rebuttal · Authors · 2025-07-29
>
> # General Response
> ----
>
> All reviewers acknowledge that our paper presents a promising approach, but also raise the question of prior mismatch, especially when analyzing real-world data. While we believe it is crucial for a new method to study synthetic data with known ground truth, we agree that analyzing real world data is important. Therefore, we have meanwhile performed an analysis of Do-PFN's performance on **six datasets based on real-world data**, with promising results that we detail next:
>
> **Known Graph:** First, we evaluate on the Amazon Sales and Law School Admissions datasets where the causal graph is known (please refer to the .zip file in the supplementary material attached to the original submission for details):
>
> ## Median MSE (CATE) - Known Graph
> |Method|Law School|Amazon Sales|
> |:------------------------|:------------------|:------------------|
> |Causal Forest DML|0.157 ± 0.056|0.136 ± 0.279|
> |Causal Forest DML (HPO)|0.519 ± 0.080|0.579 ± 0.316|
> |DML (TabPFN)|0.283 ± 0.067|**0.037 ± 0.169**|
> |Do-PFN (v1)|**0.029 ± 0.010**|**0.038 ± 0.023**|
> |Do-PFN (v1.1)|**0.067 ± 0.043**|0.209 ± 0.135|
> |Dont-PFN|**0.103 ± 0.046**|0.313 ± 0.158|
> |DragonNet|0.371 ± 0.035|0.579 ± 0.316|
> |S-Learner (HPO)|0.123 ± 0.045|0.233 ± 0.246|
> |S-Learner (TabPFN)|0.161 ± 0.051|0.568 ± 0.303|
> |TARNet|0.372 ± 0.119|0.579 ± 0.316|
> |T-Learner (HPO)|0.151 ± 0.095|**0.017 ± 0.072**|
> |T-Learner (TabPFN)|0.161 ± 0.048|0.579 ± 0.316|
> |X-Learner (HPO)|0.519 ± 0.080|0.579 ± 0.316|
> |X-Learner (TabPFN)|0.134 ± 0.047|0.579 ± 0.316|
>
> We find that Do-PFN (v1) achieves the strongest performance on Law School and is not significantly worse than the best method on Amazon Sales, a dataset with a complex mediation structure.
>
> **RealCause:** We also evaluate on four new datasets from the RealCause benchmark [5], using a version of Do-PFN (which we call v1.1) pre-trained on larger graphs with up to 60 nodes.
>
> We observe that Do-PFN-v1.1’s extended pre-training leads to strong performance among competitive baselines on three out of four RealCause datasets. We note that these datasets perfectly satisfy unconfoundedness, giving our baselines a fundamental advantage against Do-PFN. Due to space constraints we refer the Table “**Median MSE (CATE) - RealCause**” in our general response to reviewer **Fczh** for the precise MSE values.
>
> These results show Do-PFN to nevertheless be competitive with a strong set of CATE baselines, suggesting the strong transfer of Do-PFN's synthetic pre-training to complex, real-world data.
>
> # Response to Reviewer HtXo
> ----
>
> Thank you very much for your feedback and comments!
>
> > Possibility of including more variables (e.g. number of confounders in your experiment)?
>
> We are happy to provide more high-dimensional synthetic cases! We refer to a new experiment in our out-of-distribution (OOD) analysis, in which we sample 500 synthetic datasets from our prior with graphs of up to 50-nodes.
>
> ## Median MSE (CID) - OOD Graph Size
>
> |Number of Nodes|Do-PFN (v1)|Do-PFN (v1.1)|
> |:------------------|:----------------|:---------------------|
> |3-10|**0.0044 ± 0.0009**|0.0045 ± 0.0029|
> |11-20|**0.0083 ± 0.0012**|0.0091 ± 0.0026|
> |21-30|0.0107 ± 0.0012|**0.0083 ± 0.0011**|
> |31-40|0.0119 ± 0.0017|**0.0076 ± 0.0013**|
> |41-50|0.0140 ± 0.0020|**0.0051 ± 0.0018**|
>
> For larger graphs out of its prior-distribution, we observe a significant reduction in Do-PFN-v1’s performance, which decreases as the graph size goes further out of distribution. Do-PFN (v1.1), however, which was trained on graphs up to 60 nodes, achieves significantly better performance  on large graphs with 21-50 nodes. This provides evidence towards Do-PFN’s ability to scale up to more complex and high-dimensional problems.
>
> > The computational efficiency appears questionable, deploying a 7.3 million parameter model for datasets with only 5 variables raises concerns about scalability and whether such model complexity is warranted for these problem sizes.
>
> First, we show, via the performance of Do-PFN (v1.1), that using the same architecture (and number of parameters) scales up to achieve strong performance on more complex tasks, as shown in the **General Response** provided above.
>
> ## Median Train/Optimize + Inference Time (s)  - RealCause
>
> |Method|LalondeCPS|LalondePSID|IHDP|ACIC2016|
> |:-----------------------|:------------------|:------------------|:------------------|:------------------|
> |Causal Forest DML|8.075 ± 0.090|8.014 ± 0.083|6.839 ± 0.051|19.070 ± 0.239|
> |DML (TabPFN) GPU|12.376 ± 0.035|12.441 ± 0.094|11.740 ± 0.029|22.807 ± 0.271|
> |Do-PFN (v1) CPU|2.699 ± 0.047|2.324 ± 0.131|**0.699 ± 0.009**|**2.734 ± 0.179**|
> |Do-PFN (v1) GPU|**1.029 ± 0.098**|**1.311 ± 0.162**|1.270 ± 0.143|**1.060 ± 0.160**|
> |Do-PFN (v1.1) CPU|3.010 ± 0.201|2.188 ± 0.013|**0.655 ± 0.004**|4.474 ± 2.042|
> |Do-PFN (v1.1) GPU|**1.011 ± 0.002**|**0.990 ± 0.004**|0.962 ± 0.001|**1.035 ± 0.036**|
> |Dont-PFN GPU|2.916 ± 0.169|2.437 ± 0.097|**0.769 ± 0.005**|3.258 ± 0.125|
> |DragonNet GPU|67.469 ± 14.605|124.642 ± 10.753|6.791 ± 0.013|6.378 ± 0.249|
> |S-Learner (HPO) |60.160 ± 0.072|60.373 ± 0.117|60.066 ± 0.012|60.498 ± 0.206|
> |S-Learner (TabPFN) GPU|**2.001 ± 0.004**|**1.927 ± 0.012**|1.791 ± 0.002|4.694 ± 0.105|
> |T-Learner (TabPFN) GPU|3.473 ± 0.009|3.573 ± 0.168|3.309 ± 0.005|7.378 ± 0.202|
> |TARNet GPU|58.258 ± 21.274|119.821 ± 14.448|6.804 ± 0.008|6.728 ± 0.293|
> |X-Learner (HPO) |120.240 ± 0.094|120.172 ± 0.102|120.060 ± 0.007|120.201 ± 0.050|
> |X-Learner (TabPFN) GPU|13.555 ± 0.239|13.469 ± 0.216|12.886 ± 0.037|31.158 ± 2.123|
> |Shape|2500,5|1340,5|375,7|2405,50|
>
> Second, in our runtime analysis, Do-PFN (run on CPU and GPU) has a faster inference speed than all other baselines. Do-PFN outperforms other baselines because there is no conventional model-fitting, and is slightly faster than the TabPFN meta-learning approaches (when both methods use a GPU), since Do-PFN has less parameters and does not do ensembling.
>
> > [...] why does the model exhibit underconfidence in identifiable cases?
>
> This is a very interesting question, which we think can be best explained by two factors:
>
> First, the training objective for Do-PFN minimizes a **forward** KL divergence $D_{KL}[P||Q]$ between the true posterior $P$ and Do-PFN’s approximation $Q$ (refer to Proposition 1 in our paper and [6]). This encourages $Q$ to be mass-covering when it cannot approximate $P$ perfectly, meaning $Q$ will be overdispersed in cases where there is a difference between $Q$ and $P$ [2,3].
>
> Second, in the identifiable case, the posterior $P$ will be more concentrated (no need to account for unidentifiability), which means that our TabPFN-like approximation of the posterior, based on a bar-distribution [4], inevitably incurs more discretization error than in the unidentifiable case. This causes a larger KL-divergence relative to the unidentifiable case and thus, by the first point, leads to a slightly more conservative estimated distribution $Q$.
>
> We also note that underconfidence in case of approximation error is typically considered a very desirable property [3].
>
> > [...] why it performs well specifically on small datasets
>
> While performance on small datasets is a quality inherent to basically all PFNs and has been extensively empirically confirmed [4,5], from a theoretical perspective, this hasn’t been extensively explored. We believe that the best explanation is that PFNs are trained across millions of samples of small datasets, to minimize the CE loss on the test split of the dataset, which directly penalizes overfitting.
>
> > Can you run experiment[s] estimating ATE and compare with the same baselines?
>
> We have evaluated Do-PFN and an extended set of baselines (please see our first response to reviewer **Fczh** for their descriptions) on the task of ATE estimation.
>
> ## Relative Error (ATE) - Synthetic
>
> |Method|Back-Door Criterion|Confounder + Mediator|Front-Door Criterion|Observed Confounder|Observed Mediator|Unobserved Confounder|
> |:-------------------|:-----------------------|:-------------------------|:------------------------|:-----------------------|:---------------------|:-------------------------|
> |Causal Forest DML|1.256 ± 0.308|0.872 ± 0.068|1.039 ± 0.204|0.550 ± 0.105|1.000 ± 0.000|2.332 ± 0.247|
> |DML (TabPFN)|1.143 ± 0.156|1.000 ± 0.000|1.067 ± 0.118|0.853 ± 0.064|1.000 ± 0.000|1.760 ± 0.222|
> |Do-PFN (v1)|**0.369 ± 0.086**|**0.178 ± 0.038**|**0.271 ± 0.080**|**0.064 ± 0.010**|0.099 ± 0.031|**0.623 ± 0.089**|
> |Dont-PFN|2.270 ± 0.689|0.786 ± 0.053|1.063 ± 0.403|0.334 ± 0.054|0.627 ± 0.108|2.755 ± 0.441|
> |Dragon Net|4.166 ± 1.177|1.142 ± 0.093|3.340 ± 0.697|1.029 ± 0.064|1.019 ± 0.091|1.371 ± 0.174|
> |S-Learner (TabPFN)|1.572 ± 0.390|0.485 ± 0.061|1.039 ± 0.265|0.378 ± 0.067|0.285 ± 0.047|2.128 ± 0.200|
> |T-Learner (TabPFN)|9.195 ± 1.756|1.756 ± 0.264|10.692 ± 3.516|1.256 ± 0.254|**0.061 ± 0.020**|2.332 ± 0.251|
> |TARNet|3.731 ± 1.341|1.117 ± 0.072|3.706 ± 0.988|1.104 ± 0.095|1.015 ± 0.100|1.224 ± 0.155|
> |X-Learner (TabPFN)|4.603 ± 2.072|0.956 ± 0.111|2.793 ± 0.938|0.776 ± 0.160|0.139 ± 0.033|2.386 ± 0.277|
>
> We find Do-PFN to have the best performance in 5/6 case-studies, except for in the “Observed Mediator” case, in which the T and X-Learner baselines (with TabPFN v2 as a base model) is best suited to encapsulate the binary nature of this case study due to its two-model approach.
>
> ## References
>
> [1] Neal et al. "Realcause: Realistic Causal Inference Benchmarking." arXiv 2020.
>
> [2] Murphy. "Probabilistic Machine Learning: Advanced topics. MIT press." 2023.
>
> [3] McNamara, et al. "Sequential Monte Carlo for Inclusive KL Minimization in Amortized Variational Inference." AISTATS 2024.
>
> [4] Hollmann, Noah, et al. "TabPFN: A Transformer That Solves Small Tabular Classification Problems in a Second." NeurIPS 2022.
>
> [5] McElfresh, et al. "When Do Neural Nets Outperform Boosted Trees on Tabular Data?." NeurIPS 2023.

---

> > ### Comment · Reviewer_HtXo · 2025-08-01
> >
> > Thanks for answering my questions and the additional experiments supporting the strength of Do-PFNs. I am happy to increase my score.

---

### Official Review · Reviewer_Fczh · 2025-06-30

**Clarity:** 3
**Significance:** 3
**Originality:** 2
**Rating:** 4
**Confidence:** 3

**Summary:**

Do-PFN is a PFN‐based transformer for zero‐shot causal inference via meta‐training on synthetic SCMs, with a theoretical guarantee of CID optimality under its prior. Empirically, it outperforms regression and causal‐forest baselines and rivals graph‐aware methods, showing robustness and calibrated uncertainty—though its real‐world success hinges on prior fidelity.

**Questions:**

1. How does Do-PFN perform on real or semi-synthetic causal datasets (e.g., healthcare or economics benchmarks), and what evidence can you provide that your synthetic prior’s function families, noise distributions, and DAG structures cover the diversity of real-world causal mechanisms?

2. Can you quantify Do-PFN’s robustness when test data come from distributions not represented in your synthetic prior—i.e., under realistic distribution shift—and characterize how its “optimality” degrades compared to Proposition 1’s guarantees?

**Ethical Concerns:**

["NO or VERY MINOR ethics concerns only"]

**Final Justification:**

I have raised my score to 4 based on the real-world experimental results provided by the authors.

**Limitations:**

yes

**Quality:**

2

**Strengths And Weaknesses:**

### Weaknesses

- **w1:** All experiments in the paper are based on carefully designed synthetic SCMs, but none are validated on any real causal datasets (e.g., semi-synthetic or pseudo-real data in healthcare or economics). A core question is whether the function families, noise distributions, and DAG structures in the synthetic prior can cover the extremely diverse causal mechanisms encountered in real-world scenarios.
- **w2:** When the test data **are not** generated from this synthetic prior but come from other real distributions, the “optimality” guaranteed by Proposition 1 no longer holds—closely related to w1.
- **w3:** There is no systematic comparison with widely used causal-inference tools (e.g., T-learner, IPW, CFRNet, TARNet) on the same synthetic tasks.
- **w4:** The SCM is configured with a relatively small number of random nodes. It remains unclear how the method performs on high-dimensional datasets.

### Strengths

- **s1:** Proposes an end-to-end in-context causal inference approach that requires no pre-specified causal graph and does not rely on untestable no-confounding assumptions. It can automatically learn front-door and back-door adjustments, greatly simplifying the traditional causal-inference pipeline.
- **s2:** The same Do-PFN architecture can handle multiple tasks—such as CID (distribution prediction) and CATE point estimation—thereby reducing the overall complexity of the causal-inference pipeline.

---

> ### Author Rebuttal · Authors · 2025-07-29
>
> # General Response
> ----
>
> All reviewers acknowledge that our paper presents a promising approach, but also raise the question of prior mismatch, especially when analyzing real-world data. While we believe it is crucial for a new method to study synthetic data with known ground truth, we agree that analyzing real world data is important. Therefore, we have meanwhile performed an analysis of Do-PFN's performance on **six new datasets based on real-world data**, with promising results that we detail next:
>
> **Known Graph**: First, we evaluate on the Amazon Sales and Law School Admissions datasets where the causal graph is known (please refer to the supplementary material attached to the original submission as a .zip file for details). We also evaluate against six new baselines.
>
> ## Median MSE (CATE) - Known Graph
> |Method|Law School|Amazon Sales|
> |:------------------------|:------------------|:------------------|
> |Causal Forest DML|0.157 ± 0.056|0.136 ± 0.279|
> |Causal Forest DML (HPO)|0.519 ± 0.080|0.579 ± 0.316|
> |DML (TabPFN)|0.283 ± 0.067|**0.037 ± 0.169**|
> |Do-PFN (v1)|**0.029 ± 0.010**|**0.038 ± 0.023**|
> |Do-PFN (v1.1)|**0.067 ± 0.043**|0.209 ± 0.135|
> |Dont-PFN|**0.103 ± 0.046**|0.313 ± 0.158|
> |DragonNet|0.371 ± 0.035|0.579 ± 0.316|
> |S-Learner (HPO)|0.123 ± 0.045|0.233 ± 0.246|
> |S-Learner (TabPFN)|0.161 ± 0.051|0.568 ± 0.303|
> |TARNet|0.372 ± 0.119|0.579 ± 0.316|
> |T-Learner (HPO)|0.151 ± 0.095|**0.017 ± 0.072**|
> |T-Learner (TabPFN)|0.161 ± 0.048|0.579 ± 0.316|
> |X-Learner (HPO)|0.519 ± 0.080|0.579 ± 0.316|
> |X-Learner (TabPFN)|0.134 ± 0.047|0.579 ± 0.316|
>
> We find that Do-PFN (v1) achieves the strongest performance on Law School and is not significantly worse than the best method on Amazon Sales, a dataset with a complex mediation structure.
>
> **RealCause:** Furthermore, we have pre-trained a version of Do-PFN (which we call v1.1) on larger graphs with up to 60 nodes, in order to evaluate on datasets from the RealCause benchmark [5].
>
> ##  Median MSE (CATE) - RealCause
> |Method|LalondeCPS|LalondePSID|IHDP|ACIC2016|
> |:-------------------|:------------------|:------------------|:------------------|:------------------|
> |Causal Forest DML|0.114 ± 0.014|0.016 ± 0.002|0.024 ± 0.001|0.005 ± 0.001|
> |DML (TabPFN)|0.067 ± 0.009|0.013 ± 0.001|0.025 ± 0.002|0.008 ± 0.004|
> |Do-PFN (v1)|0.067 ± 0.006|0.022 ± 0.003|0.031 ± 0.002|0.033 ± 0.004|
> |Do-PFN (v1.1)|**0.039 ± 0.004**|**0.008 ± 0.001**|0.021 ± 0.001|0.019 ± 0.011|
> |Dont-PFN|0.049 ± 0.004|0.011 ± 0.001|0.032 ± 0.001|0.036 ± 0.003|
> |DragonNet|0.064 ± 0.006|0.020 ± 0.003|0.543 ± 0.154|0.097 ± 0.035|
> |S-Learner (HPO)|0.068 ± 0.010|0.015 ± 0.001|0.022 ± 0.002|0.003 ± 0.001|
> |S-Learner (TabPFN)|0.053 ± 0.007|0.013 ± 0.003|**0.019 ± 0.001**|**0.002 ± 0.001**|
> |T-Learner (HPO)|**0.044 ± 0.003**|**0.007 ± 0.001**|0.023 ± 0.002|0.002 ± 0.001|
> |T-Learner (TabPFN)|**0.042 ± 0.002**|**0.007 ± 0.001**|**0.020 ± 0.001**|**0.002 ± 0.001**|
> |TARNet|0.064 ± 0.006|0.020 ± 0.003|0.517 ± 0.147|0.092 ± 0.032|
> |X-Learner (HPO)|0.069 ± 0.008|0.020 ± 0.003|0.155 ± 0.033|0.048 ± 0.016|
> |X-Learner (TabPFN)|0.083 ± 0.010|0.015 ± 0.005|**0.019 ± 0.001**|**0.001 ± 0.000**|
>
> We observe that Do-PFN-v1.1’s extended pre-training leads to strong performance among competitive baselines on three out of four RealCause datasets. We note that these datasets perfectly satisfy unconfoundedness, giving our baselines a fundamental advantage against Do-PFN.
>
> These results show Do-PFN to nevertheless be competitive with a strong set of CATE baselines, suggesting the strong transfer of Do-PFN's synthetic pre-training to complex, real-world data.
>
> # Response to Reviewer Fczh
> ----
> Thank you for your constructive feedback!
>
> > There is no systematic comparison with widely used causal-inference tools (e.g., T-learner, IPW, CFRNet, TARNet) on the same synthetic tasks.
>
> In response, we have evaluated Do-PFN on our synthetic case studies against a strong set of meta-learning and double machine learning (DML) baselines with TabPFN (v2) as a base learner, following [1], who use only meta-learners and [2], who show TabPFN to be a strong base learner for meta-learning CATE estimators. We also include TARNet [3] and DragonNet [4] implemented in the CATENets library.
>
> ## Median MSE (CATE) - Synthetic
> |Method|Back-Door Criterion|Confounder + Mediator|Front-Door Criterion|Observed Confounder|Observed Mediator|Unobserved Confounder|
> |:-------------------|:-----------------------|:-------------------------|:------------------------|:-----------------------|:---------------------|:-------------------------|
> |Causal Forest DML|0.820 ± 0.329|1.508 ± 0.873|0.851 ± 1.035|0.278 ± 0.059|200.944 ± 109.848|4.325 ± 0.754|
> |DML (TabPFN)|0.667 ± 0.148|0.717 ± 0.277|1.093 ± 0.538|0.435 ± 0.086|258.944 ± 210.589|2.161 ± 0.821|
> |Do-PFN (v1)|**0.159 ± 0.108**|**0.223 ± 0.154**|**0.186 ± 0.104**|**0.017 ± 0.014**|10.900 ± 9.947|**0.291 ± 0.074**|
> |Dont-PFN|3.529 ± 1.817|1.923 ± 0.750|2.884 ± 7.457|0.475 ± 0.247|313.138 ± 383.839|5.116 ± 0.887|
> |DragonNet|35.127 ± 44.199|18.362 ± 10.619|50.792 ± 47.052|6.123 ± 1.598|3817.618 ± 2773.768|7.526 ± 2.265|
> |S-Learner (TabPFN)|1.343 ± 0.765|0.542 ± 0.188|1.205 ± 0.963|0.329 ± 0.131|14.988 ± 11.032|3.760 ± 0.896|
> |T-Learner (TabPFN)|26.049 ± 10.711|6.592 ± 3.319|55.158 ± 59.264|3.034 ± 0.840|**3.434 ± 2.537**|4.430 ± 0.500|
> |TARNet|35.756 ± 26.782|19.177 ± 8.814|52.404 ± 48.647|7.172 ± 2.349|4229.619 ± 3540.554|8.003 ± 2.861|
> |X-Learner (TabPFN)|6.280 ± 3.720|3.831 ± 2.562|9.344 ± 6.165|0.921 ± 0.465|5.175 ± 4.380|4.141 ± 0.602|
>
>
> Across our six synthetic case studies, we find that Do-PFN achieves the best performance on all cases except for “Observed Mediator” in which the two-model T- and X-learner strategies are best able to capture this highly binary treatment effect, due to the treatment being the root cause.
>
> > Can you quantify Do-PFN’s robustness when test data come from distributions not represented in your synthetic prior [..]?
>
> Robustness to prior mismatch is an important topic for PFNs in general, but no theoretical results of the form suggested by the reviewer exist (also not for the non-causal TabPFN - this is a very hard problem). However, in addition to the strong real-world performance we observe above, we also perform several ablation studies regarding distribution mismatch where we vary core characteristics of the SCM-generated data Do-PFN is evaluated on.
>
> ## Median MSE - Out of Distribution
>
> |OOD Type|Do-PFN (v1)|% Change|
> |:--------------------|:--------------------|:-----------|
> |In Distribution|**0.0053 ± 0.0008**|0.0%|
> |ELU Non-Linearity|0.0033 ± 0.0004|-38.2%|
> |Post Non-Linearity|0.0057 ± 0.0006|7.0%|
> |Sin Non-Linearity|0.0066 ± 0.0007|24.6%|
> |Student-t Noise|0.0050 ± 0.0006|-4.6%|
> |Laplacian Noise|0.0060 ± 0.0008|13.3%|
> |Gumbel Noise|0.0059 ± 0.0007|10.9%|
> |High Gaussian Noise|0.0212 ± 0.0006|301.4%|
>
>
> We find that Do-PFN’s performance is robust to different noise distributions, and using post-non-linearities (instead of adding the noise after applying a nonlinearity). Using different nonlinearities can even improve performance while increasing the variance of the Gaussian noise drastically hurts performance.
>
> While not a general form of Proposition 1, one can make the following argument regarding the mismatch between training and testing data. For the posterior $p(y|x, do(t), D)$ denote $x = (x,t, D)$, such that under abuse of notation $p(y|x, do(t), D) = p(y|x)$. Denote Do-PFN’s error as a function of $p$ as $E_{p(x)} D_{KL}(p(y|x)||q(y|x)) =: e(p)$. Assume that $e \in F$ for some function class $F$. Now let the testing data come from a different distribution $p’$ and write $e’(p) = E_{p’(x)} D_{KL}(p(y|x)||q(y|x))$.
> By definition of the integral probability metric $d_F$ wrt. $F$, we get:
> $|e(p) - e’(p)| \leq d_F(p, p’) $. In case $F$ is the space of 1-Lipschitz functions, $d_F(p, p’) = W_1$ by the dual representation of the 1-Wasserstein distance $W_1$.
>
> Intuitively, when the loss of Do-PFN is well behaved and the real-world data distribution $p’$ has a difference of $C$ to the synthetic training distribution $p$, the same constant $C$ bounds how much the loss increases when taking inputs from $p’$ instead of $p$.
>
> Having said that, while the (surprisingly) strong generalization of PFNs from synthetic training data to real-world evaluation datasets has been extensively empirically confirmed, the theory behind this phenomenon is still developing [6].
>
> > It remains unclear how the method performs on high-dimensional datasets.
>
> We refer to a new experiment, in which we sample 500 synthetic datasets from our prior with graphs of up to 50 nodes.
>
> ### Median MSE (CID) - OOD Graph Size
> |Number of Nodes|Do-PFN (v1)|Do-PFN (v1.1)|
> |:------------------|:----------------|:---------------------|
> |3-10|**0.0044 ± 0.0009**|0.0045 ± 0.0029|
> |11-20|**0.0083 ± 0.0012**|0.0091 ± 0.0026|
> |21-30|0.0107 ± 0.0012|**0.0083 ± 0.0011**|
> |31-40|0.0119 ± 0.0017|**0.0076 ± 0.0013**|
> |41-50|0.0140 ± 0.0020|**0.0051 ± 0.0018**|
>
> For larger graphs out of its prior-distribution, we observe a significant reduction in Do-PFN-v1’s performance, which decreases as the graph size goes further out of distribution. Do-PFN (v1.1), however, which was trained on graph sizes up to 60 nodes, achieves significantly better performance on large graphs with 21-50 nodes. This provides evidence towards Do-PFN’s ability to scale up to more complex and high-dimensional problems.
>
> ## References
> [1] Vanderschueren et al. "AutoCATE: End-to-End, Automated Treatment Effect Estimation." ICML 2025.
>
> [2] Zhang et al. "TabPFN: One Model to Rule Them All?." arXiv 2025.
>
> [3] Shalit et al. "Estimating individual treatment effect: generalization bounds and algorithms." ICML 2017.
>
> [4] Shi et al. "Adapting neural networks for the estimation of treatment effects." NeurIPS 2019.
>
> [5] Neal et al. "Realcause: Realistic causal inference benchmarking." arXiv 2020.
>
> [6] Nagler. "Statistical foundations of prior-data fitted networks." ICML 2023.

---

> > ### Comment · Reviewer_Fczh · 2025-08-04
> >
> > Thank you for the additional real-world experiments. This addresses some of my concerns. I hope the authors will incorporate this into the revised paper, and I will accordingly raise my score.

---

### Note · Authors · 2025-08-13

We would like to thank all reviewers for the constructive feedback and for the opportunity to clarify and extend our work during the discussion phase. We are pleased that, following our rebuttal, all reviewers acknowledged the novelty and promise of Do-PFN, and that all except one, who already provided a “5: Accept”, agreed to increase their scores.

In response to the main concerns on **prior mismatch** and **real-world applicability**, we conducted extensive new experiments on six datasets derived from real-world data (Amazon Sales, Law School Admissions, and four RealCause benchmarks), showing that Do-PFN achieves competitive performance compared to a strong set of CATE estimation baselines. These results, together with our OOD ablations (varying graph size, nonlinearities, and noise), demonstrate strong robustness and transfer.

Regarding **scalability**, we evaluated Do-PFN on graphs up to 50–60 nodes, showing strong performance when pre-training covers this range. Our runtime analysis confirms that Do-PFN offers competitive inference speed to PFN and tree-based baselines.

On **model behavior**, we provided a theoretical breakdown of uncertainty sources (aleatoric, unidentifiability, epistemic) and explained why forward-KL training in combination with our bar-distribution leads to slight underconfidence in identifiable cases.

Finally, we clarified differences to related work, highlighting distinct tasks as well as pre-training and evaluation setups, and we ran additional ATE experiments confirming strong performance across diverse scenarios.

We believe these additions successfully addressed the reviewer’s concerns and further supported Do-PFN’s contribution, and are happy to include all changes and all additional experiments discussed above into the revised version of the manuscript. We sincerely thank the reviewers for their input, helping to further improve our work.

---

### Decision · Program_Chairs · 2025-09-17

**Decision:**

Accept (spotlight)

**Comment:**

This paper presents a new approach named Do-PFN for zero‐shot causal inference via meta‐training, which extends the prior-data fitted networks (PFNs) to dealing with treatment effect estimation task on tabular data. Extensive experiments with ablation studies demonstrate the effectiveness of this approach.

Reviewers recognized the novelty and technical contributions of this work. This paper provides an innovative approach for applying PFNs to causal inference. Theoretical analysis provides valuable insights on why the proposed method works. Moreover, the experimental results and analysis are comprehensive and convincing.

Meanwhile, reviewers found that the paper has some limitations, such as lack of evaluations on real-world datasets, missing baselines, computational efficiency, etc. The authors have provided detailed responses with additional results that help address the concerns from reviewers.

Overall, this paper is a solid work that extends PFNs to dealing with causal inference tasks. Considering the novelty and technical contributions of this work, I recommend it for Spotlight presentation.